# Body size estimation from isolated fossil bones reveals deep time evolutionary trends in North American lizards

**Sara J. ElShafie**  *

Department of Integrative Biology and Museum of Paleontology, University of California, Berkeley, California, United States of America

* selshafie@berkeley.edu

**Data Availability Statement:** All relevant data are within the paper and its Supporting Information.

**Funding:** This work was supported by the Anthony Barnosky Graduate Student Research Fund and the Doris O. and Samuel P. Welles Research Fund

## Abstract

Lizards play vital roles in extant ecosystems. However, their roles in extinct ecosystems are poorly understood because the fossil record of lizards consists mostly of isolated bones. This makes it difficult to document changes in lizard morphology and body size over time, which is essential for studies of lizard paleoecology and evolution. It is also difficult to compare available fossil lizard data with existing sources of extant lizard data because extant studies rarely measure individual bones. Furthermore, no previous study has regressed measurements of individual bones to body length across crown lizard groups, nor tested those regressions on fossil skeletons. An extensive dataset of individual bone measurements from extant lizards across crown taxonomic groups is here employed to develop novel methods for estimating lizard body size from isolated fossil elements. These methods were applied to a comparably large dataset of fossil lizard specimens from the robust Paleogene record (66–23 Ma) of the Western Interior of North America. This study tests the hypothesis that anatomical proportions have been conserved within higher-level crown lizard groups since the Paleogene and can therefore be used to reconstruct snout-vent length (SVL) and mass for fossil specimens referred to the same groups. Individual bones demonstrated strong correlation with SVL in extant as well as fossil lizard specimens ($R^2 \geq 0.69$). Equations for mass estimation from individual bones were derived from the SVL regressions using published equations for calculating lizard body mass from SVL. The resulting body size estimates from regression equations for the entire fossil dataset revealed that lizards reached greatest maximum body size in the middle Paleogene, with the largest size class dominated by anguid lizards that exceeded 1 meter in SVL and 1.5 kg in body mass. Maximum body size decreased to under 400 mm and below 1.5 kg in the late Paleogene. No association was found between changes in maximum lizard body size and marine isotope proxies of global temperature through the Paleogene. This is the first study to investigate body size evolution across lizard clades over a deep time interval and for a large geographic region. The proposed methods can be used to generate body size regressions and provide estimates of body size for isolated lizard bones referred to any crown group.

(University of California Museum of Paleontology – ucmp.berkeley.edu), the Department of Integrative Biology at the University of California, Berkeley (ib. berkeley.edu), the Mary R. Dawson Award and the Albert E. Wood Award (Society for Vertebrate Paleontology – vertpaleo.org), the Charles A. and June R.P. Ross Research Award (Geological Society of America – www.geosociety.org), the Evolving Earth Foundation Graduate Student Research Grant (www.evolvingearth.org), the Burke Museum Vertebrate Paleontology Collections Study Grant (www.burkemuseum.org), the Sigma Xi Grant in Aid of Research (www.sigmaxi.org), the Friends of the Nebraska State Museum (friendsofthemuseum.org), the Nebraska Geological Society (www.nebraskageologicalsociety.org), the Department of Earth and Atmospheric Sciences at the University of Nebraska, Lincoln (eas.unl.edu), the Sakana Foundation, and the Uplands Foundation. The funders had no role in study design, data collection and analysis, decision to publish, or preparation of the manuscript.

**Competing interests:** The author has declared that no competing interests exist.

**Abbreviations:** AMNH, American Museum of Natural History, New York, NY; CAS, California Academy of Sciences, San Francisco, CA; CJB, Christopher J. BellUniversity of Texas at Austin, Austin, TX; CM, Carnegie Museum of Natural History, Pittsburgh, PA; DMNH, Denver Museum of Natural History (Denver Museum of Nature and Science), Denver, CO; FLMNH, Florida Museum of Natural History, Gainesville, FL; FMNH, Field Museum of Natural History, Chicago, IL; FOBU, Fossil Butte National Monument, Kemmerer, WY; GCVP, Georgia College Vertebrate Paleontology, Milledgeville, GA; LACM, Natural History Museum of Los Angeles County, Los Angeles, CA; MCZ, Museum of Comparative Zoology, Cambridge, MA; MSU, Michigan State University, Lansing, MI; MVZ, Museum of Vertebrate Zoology, Berkeley, CA; NMMNH, New Mexico Museum of Natural History, Albuquerque, NM; PTRM, Pioneer Trails Regional Museum, Bowman, ND; RAM, Raymond M. Alfe Museum of Paleontology, Claremont, CA; SMM, Science Museum of Minnesota, Saint Paul, MN; TxVP, Texas Vertebrate Paleontology Collection, The University of Texas, Austin, TX; UCM, University of Colorado Museum, Boulder, CO; UCMP, University of California Museum of Paleontology, Berkeley, CA; UMMP, University of Michigan Museum of Paleontology, Ann Arbor, MI; UNSM, University of Nebraska State Museum, Lincoln, NE; USNM, United States National Museum (Smithsonian National Museum of

## Introduction

Body size influences every aspect of a vertebrate's physiology and life history [1–4]. It is therefore essential to estimate body size for fossil vertebrate taxa in order to understand the ecological and evolutionary context of any vertebrate group [5–10]. Lizards are a particularly important group of vertebrates to investigate in both extant and extinct ecosystems because they exhibit high taxonomic and morphological diversity and occupy a wide range of ecological roles [4,11–15]. For example, lizards contribute significant tetrapod species richness in some environments, especially deserts. They can maintain high population densities in small areas and have high dispersal abilities [16–20]. Lizards also regulate populations of insects, rodents, and other prey on which they feed, and they are an important food source for many more animal species [4,12,13,21].

The lizard fossil record consists mostly of isolated elements that can be difficult to identify to the level of genus or even family. Therefore, body size for fossil lizards must be inferred primarily from isolated bones [22]. To further complicate matters, anatomical datasets of extant lizards rarely include measurements of individual cranial or limb bones for comparison. Most relevant previous studies only recorded measurements of head length and snout-vent length (SVL) across extant lizard groups [3,4,11,12,20,23–27]. Hence, no prior study has reconstructed body size in lizards on evolutionary time scales across higher-level taxonomic groups. This study presents an extensive reference dataset of individual anatomical element measurements taken across several crown group lizard lineages and develops methods for using those data to estimate body size in fossil lizards.

It is not unambiguously clear which variable is the best proxy for body size in fossil vertebrates. Some previous studies estimated mass as a proxy for body size in fossil crocodyliforms using scaling relationships in extant taxa between body mass and femoral length [28], femoral circumference [28], or head width [29], since these anatomical measurements are commonly available in the crocodyliform fossil record. Body mass has also been estimated for dinosaurs [7] and fossil mammals [6,30,31] using similar methods. However, body mass is only weakly correlated with most individual bones in many fossil vertebrates; thus, such methods can be difficult to apply across a fossil record of varied isolated bones [6].

For fossil reptiles, body length is easier to estimate from individual bones than body mass because length correlates tightly with other anatomical measurements. Body length has been shown to correlate with femur length ([28]), vertebral element width ([32]), head length ([33]), head width ([29]), and specific cranial bones such as dentary length [9,34] in extant lizards, snakes, and crocodylians ($R^2 \geq 0.78$ in all cited examples). Mandible length has also been used as a direct proxy for body size in fossil lizards [22,35].

SVL, measured from the tip of the snout to the cloacal opening at the base of the tail (Fig 1), is often the preferred measure of body length in extant lizard studies rather than total body length [23,34]. This is because many lizards can autotomize and even regenerate their tails [36–40], making lizard tail measurements highly variable. Total body length can also be difficult to reconstruct for dry skeletonized specimens, which are often disarticulated. Measurements of SVL are more feasible to obtain from wet preserved or dry skeletonized extant lizard specimens. Maximum SVL strongly correlates with mean adult SVL and SVL at sexual maturity in extant lizards [3,41], so maximum SVL is a good metric of mean adult body size for a population of lizards sampled from the fossil record. Furthermore, once SVL is estimated, lizard body mass can be calculated from family-specific SVL-to-mass equations [25,42,43].

For this study, lizards were sampled from the Paleogene record (66–23 Ma) in the Western Interior of North America, specifically, from localities concentrated in the United States (Fig 2). The Paleogene is an interesting period to study because it spans significant warming and

Natural History), Washington, DC; UW, University of Wyoming Geological Museum, Laramie, WY; UWBM, University of Washington Burke Museum, Seattle, WA; YPM, Yale Peabody Museum, New Haven, CT.

cooling events [44,45] and chronicles the radiations of both mammals [46–56] and later squamates [57–61] following the End-Cretaceous Mass Extinction. This timeframe was used here as a study system because it has a prolonged and robust fossil record in the Western Interior, preserving many diverse and ecologically important extant lizard clades that still occur in the area. These include anguids and pleurodontan iguanians (e.g., iguanids and dactyloids), as well as clades that became locally extinct in the region during the Paleogene, such as varanids and shinisaurids. All the sampled Paleogene specimens, many identified to the level of genus or even species, were stored in natural history museum collections in the United States.

This study tested the hypothesis that proportions of individual bone lengths to SVL are conserved within each of eight crown group lizard lineages (exclusive of snakes) represented in the Paleogene fossil record of the Western Interior, and that these proportions can therefore be used to predict SVLs for fossil lizard specimens. For each crown group investigated, individual cranial and limb bones were regressed onto SVL, and the resulting equations were validated using complete skeletons of both extant and fossil lizards. The SVL regressions were also used to generate novel equations for mass estimation from individual bones using published equations for calculating extant lizard body mass from SVL [25,42,43]. The equations were then applied to reconstruct body size for crown group lizards sampled from across the U.S. Western Interior through the Paleogene. This study provides the first opportunity to investigate modes of body size evolution across lizard diversity over a prolonged geologic interval and spanning a continental interior.

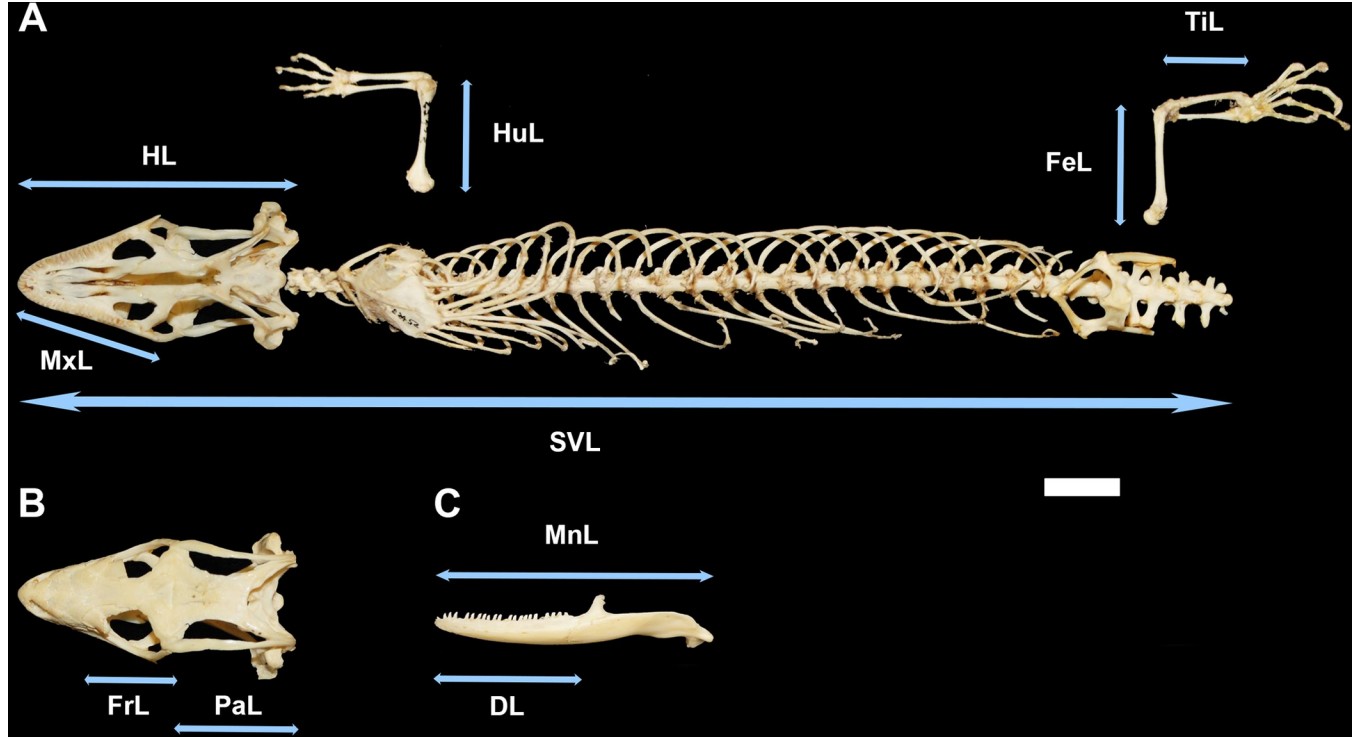

**Fig 1. Anatomical measurements of lizard skeleton.** A) Skull, vertebral series, left forelimb and left hind limb in ventral view. B) Skull in dorsal view. C) Left mandible in lateral view. A-C are oriented with anterior to the left. Measurements are indicated by arrows. Snout-vent length (SVL) was measured from the tip of the snout at the premaxilla to the posterior edge of the centrum on the second caudal vertebra, which is the approximate position of the middle of the cloaca (vent). Other measurement acronyms: DL = dentary length, FeL = femur length, FrL = frontal length, HL = head length, HuL = humerus length, MnL = mandible length, MxL = maxilla length, PaL = parietal length, TiL = tibia length. Scale bar = 1 cm. Specimen pictured is *Gerrhonotus infernalis* (Anguidae; FMNH 22452).

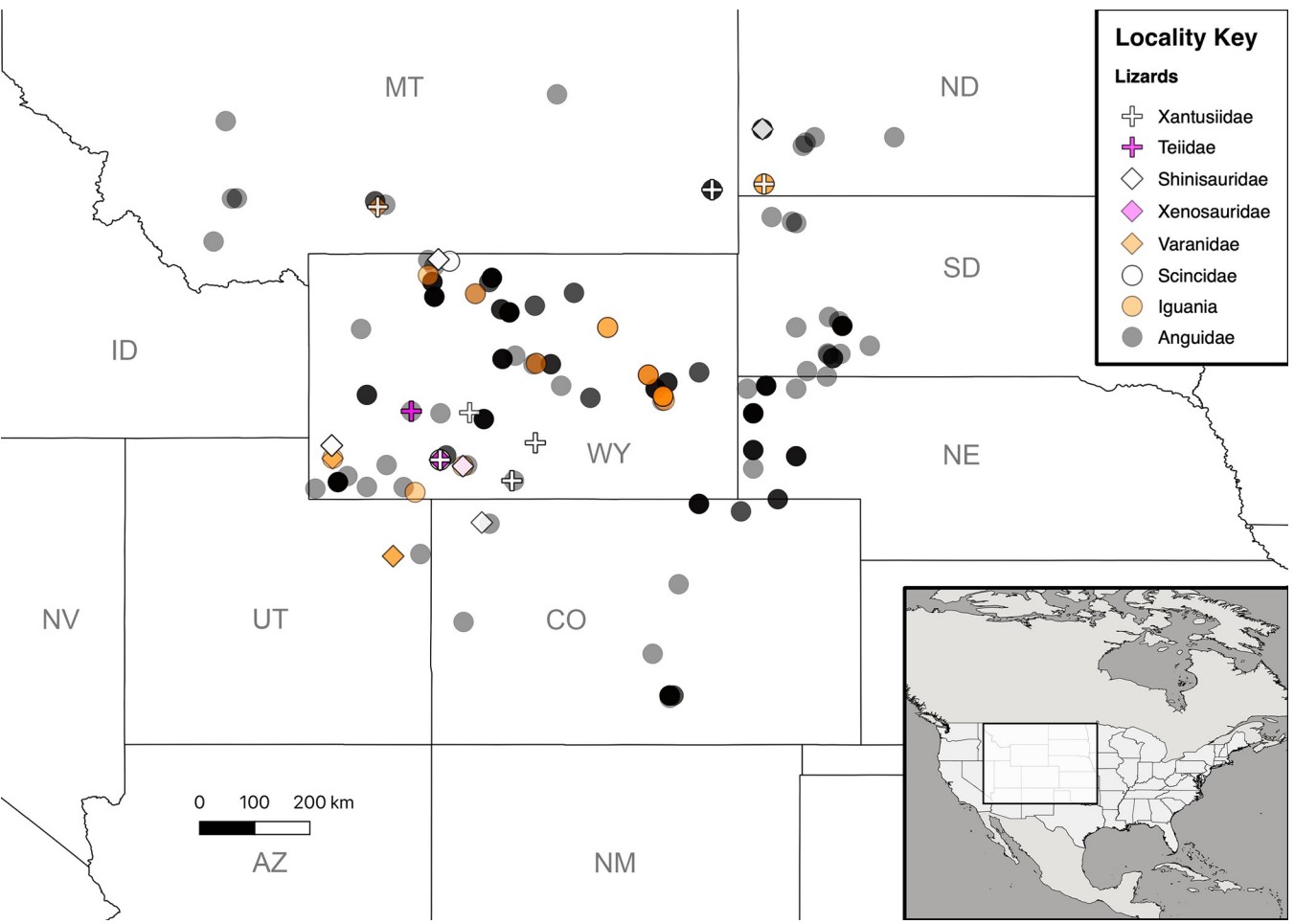

**Fig 2. Map showing all localities for fossil lizard data from across the Western Interior of North America through the Paleogene.** Featured area is highlighted in inset map. Data points represent only specimens for which the locality could be georeferenced. Some taxonomic groups occurring with others in a particular locality may not be visible. Base maps were made with Natural Earth and are reprinted from Natural Earth under a CC Public Domain license, original copyright 2009. All other data contained were collected by the author. Maps were generated using QGIS-LTR 3.22.6 (download.qgis.org). The figure created is under the CC-BY 4.0 license, as determined by the authors of QGIS.

## Materials & methods

### Data collection

The data for this study were collected from 26 different natural history museum collections across the United States (see "Institutional Abbreviations"). These include anatomical measurements from a total of 283 fossil lizard specimens, with 44 recognized genera and at least 33 recognized species, from the Paleogene record of the U.S. Western Interior (Table 1 and S1 Dataset). Only fossil lizard specimens with complete cranial bones or associated limb bones were sampled (e.g., Fig 3), for the following reasons: 1) the lizard fossil record is mostly comprised of cranial bones and, fortunately, most extant phylogenetic morphological characters for lizards are found in cranial bones [62,63], making it possible to determine at least coarse taxonomic identifications (family or higher) for fossil lizard specimens from cranial material [63]; 2) historical fossil identification may lack fidelity at the genus or species level [64–66]; 3) cranial and limb bones can be used to estimate SVL in lizards [9,34]. A few measured specimens included complete skulls (Fig 3A and 3I) or even skeletons (Figs 4A and 5). All fossil

**Table 1. Summary of datasets.** For the extant species totals, subspecies were counted separately. Fossil species totals do not include specimens that could not be identified to a specific genus or species (e.g., Anguidae indet.). For example, all three fossil teiid specimens sampled were identified as only "Teiidae indet".

| Taxonomic group | # Fossil specimens sampled | # Fossil genera sampled | # Fossil species sampled | # Extant specimens sampled | # Extant genera sampled | # Extant species sampled |
|---|---|---|---|---|---|---|
| Anguidae | 218 | 16 | 17 | 66 | 4 | 17 |
| Varanidae | 6 | 1 | 1 | 47 | 1 | 17 |
| Teiidae | 3 | 0 | 0 | 33 | 5 | 11 |
| Scincidae | 4 | 3 | 1 | 28 | 8 | 19 |
| Xantusiidae | 19 | 2 | 2 | 20 | 1 | 3 |
| Xenosauridae | 7 | 4 | 3 | 17 | 1 | 3 |
| Iguania | 19 | 8 | 6 | 55 | 9 | 16 |
| Shinisauridae | 7 | 3 | 3 | N/A | N/A | N/A |
| Total | 283 | 37 | 33 | 266 | 29 | 86 |

specimens were referred to crown groups (family-level or higher) based on specimen label identifications as well as taxonomic information from the Paleobiology Database [67] and current literature.

To compare the fossil lizard specimens with extant congeners, 266 extant specimens were also measured (S2 Dataset), comprising 30 genera and 86 species (Table 1) from the same crown group lineages represented in the fossil dataset. These included seven extant families: Anguidae (e.g., alligator lizards and galliwasps, Fig 3C), Scincidae (skinks), Shinisauridae (crocodile lizards), Teiidae (e.g., whip-tail lizards), Varanidae (monitor lizards), Xantusiidae (night lizards), and Xenosauridae (knob-scaled lizards). Legless anguids and scincids were omitted because proportions of cranial elements to SVL differ between limbed taxa and serpentiform taxa in extant squamate groups [22] and because all the extinct taxa in the fossil dataset presented here are inferred to have had limbs based on associated limb material or assignment to extant groups with limbs. The pleurodontan iguanian families represented in the fossil dataset (e.g., Iguanidae and Polychrotidae (Fig 5B)) were treated as a collective group, "Iguania," in both datasets. Phylogenetic relationships among iguanian families are poorly resolved, but the group "Iguania" is supported by both morphological and molecular analyses [68–71].

In order to compare individual cranial and limb measurements with SVL measurements, extant data collection in this study focused on dry skeletonized specimens with reasonably complete skulls and vertebral series (Fig 1 and S2 Dataset). A total sample size of at least n = 17 specimens was collected for each of the eight major taxonomic groups (it was difficult to find complete specimens of Xenosauridae, so that sample size was the lowest), with 1–21 individuals sampled per species.

Extant species were selected for sampling to encompass the full range of extant body size diversity within a given lineage as much as possible, with the exception of extreme outliers. However, some outlier specimens were used to further test the predictive power of regressions for groups that included exceptionally large individuals in the fossil dataset, Anguidae and Varanidae (S1 and S2 Tables).

All measurements were taken with digital calipers (Mitutoyo 150 mm) to the nearest 0.1 mm for individual bones and the nearest 1 mm for SVL. For specimens with an SVL measuring > 150 mm, or for specimens that were preserved in a curved position, a tape measure was used to measure SVL to the nearest 1 mm. SVL was measured on skeletons as the length from the tip of the snout to the posterior centrum of the second caudal vertebra, which is the approximate position of the middle of the cloaca (vent; see Fig 1). The position of the cloaca on skeletonized specimens was determined by aligning the pelvis of a wet specimen with

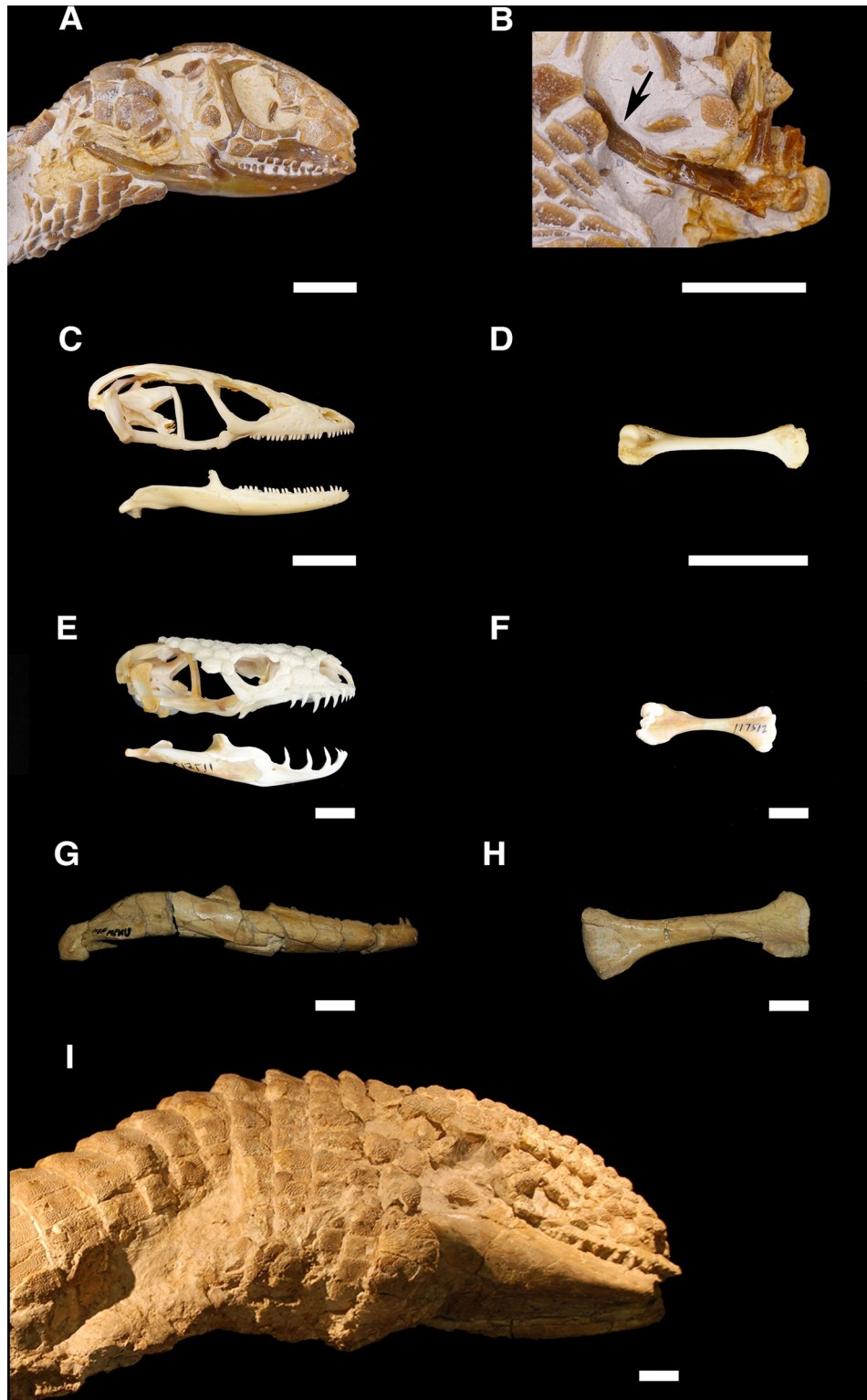

**Fig 3. Paleogene anguimorph lizards from the Western Interior of North America and extant morphological analogues.** (A) *Peltosaurus granulosus†* (Anguidae) skull in right lateral view, and B) right humerus in medial view (AMNH 42913), indicated by arrow, Orellan (33.9–31.8 Ma). (C) *Gerrhonotus infernalis* (Anguidae) skull and right mandible in right lateral view and (D) left humerus in medial view (FMNH 22452), extant. (E) *Heloderma suspectum* (Helodermatidae) skull and left mandible in left lateral view, reflected for continuity, and (F) left humerus in medial

view (UCMP 117512), extant. (G) *Helodermoides tuberculatus*[†] (Anguidae) left mandible in lateral view, reflected for continuity, and (H) right humerus in medial view (UNSM 4511), Chadronian (37.0–33.9 Ma). (I) *Helodermoides tuberculatus*[†] skull, mandibles, and cervical osteoderms in right lateral view (USNM V 13869), Chadronian. A-I are oriented with anterior to the right. Scale bar = 1 cm. USNM V 13869 (I) image courtesy of the Smithsonian Institution.

that of a dry skeletonized congener specimen of comparable size. Whenever a dried skin was included with a skeletonized specimen, it was used to verify the SVL measurement from the vertebral series. Skull length was measured as the length from the tip of the snout to the most posterior point of the skull (either the occipital condyle or the posterior tip of the supratemporal process; Fig 1). If a cranial bone had bilateral asymmetry (e.g., if on a parietal, one supratemporal process was longer than the other), the longer side was taken to represent the maximum measurement. Each cranial bone was measured from the most anterior to the most posterior point (e.g., on the dentary, from the anterior tip of the ramus to the posterior tip of the retroarticular process). For long bones, only complete bones with epiphyses included were measured.

When it was not possible to access a fossil specimen in person (which was the case for the UMMP specimens and a few specimens measured from figures in literature), measurements were taken from digital photographs with a scale bar in standard orientation using the open access software ImageJ (available online at: https://imagej.nih.gov/ij/download.html). Measurements taken from photographs are indicated in the "Notes" column in S1 Dataset.

No permits were required for the described study, which complied with all relevant regulations.

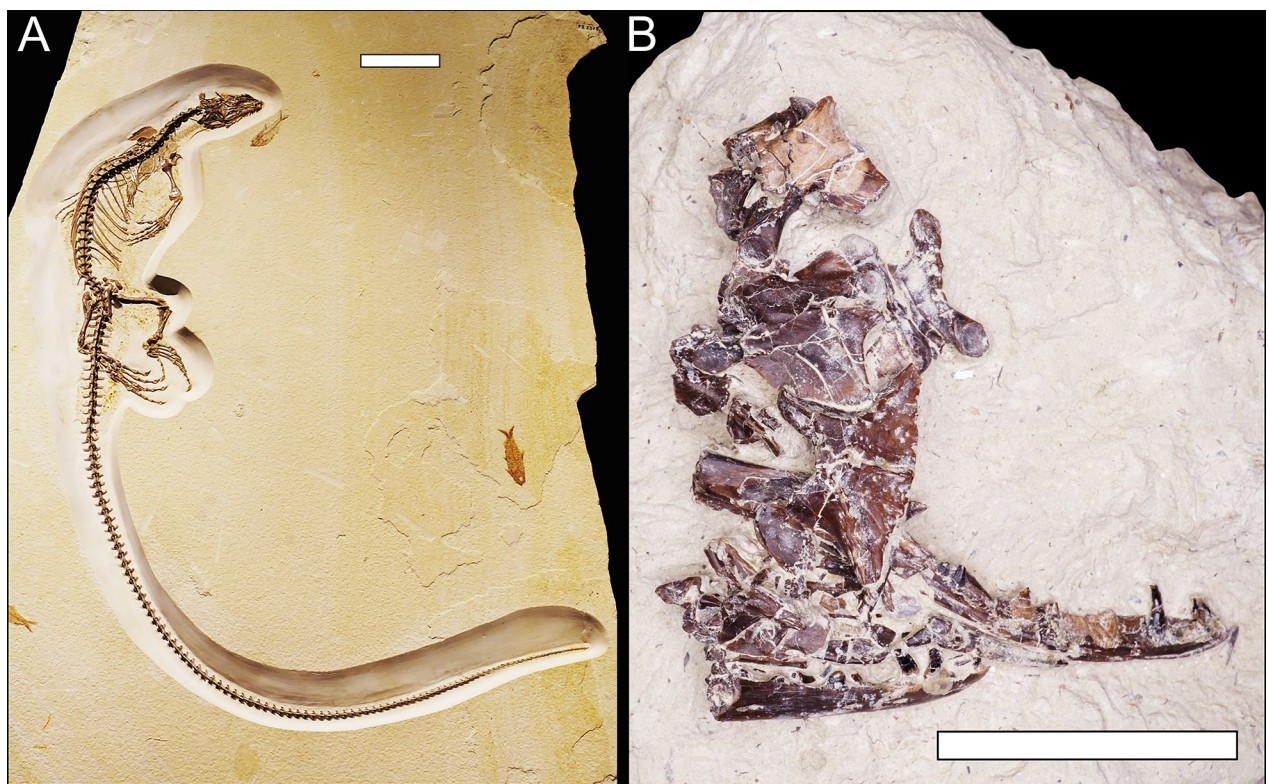

**Fig 4. Paleogene varanids from the Western Interior of North America.** (A) *Saniwa ensidens*[†] skeleton (FMNH PR 2378), Wasatchian (54.9–50.5 Ma). Scale bar = 10 cm. (B) *Saniwa sp.*[†] dentaries, vertebrae, and fragments (DMNH EPV.34588), Bridgerian (50.5–46.2 Ma). Scale bar = 3 cm.

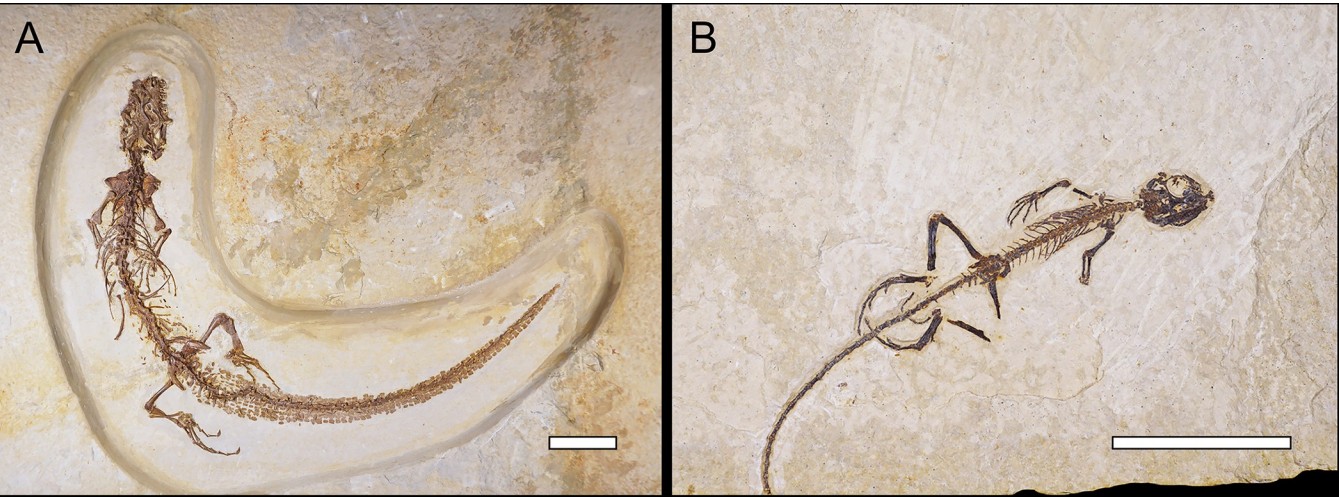

**Fig 5. Other complete Paleogene lizard skeletons from the Western Interior of North America.** (A) *Bahndwivici ammoskius*[†] (Shinisauridae) skeleton (FMNH PR 2260), Wasatchian (54.9–50.5 Ma). Scale bar = 3 cm. (B) *Afairiguana avius*[†] (Iguania; Polychrotidae) skeleton (FMNH PR 2379), Wasatchian. Scale bar = 3 cm.

## Generation and testing of regressions for body length

This study regressed anatomical elements (the independent variables)–including head length, individual cranial bone length (dentary, mandible, maxilla, frontal, parietal), and limb bone length (humerus, femur, tibia)–to SVL for each extant lizard group in the dataset presented (S1 Fig and S1–S7 Tables). Regressions were only generated for the anatomical elements for which more than one specimen was available for a given lizard group in the fossil dataset (S1–S7 Tables). For example, only one fossil varanid specimen with complete femora was found in collections (Fig 4A), so a regression for varanid femur length to SVL was not generated (S2 Table).

Reduced major axis (RMA) bivariate linear regressions were used because they accommodate error in both the dependent and independent variable measurements [72]. Before the analyses, all the measurements were transformed using the natural log (LN). This method is commonly applied in analyses of fossil vertebrate body size, especially to minimize the effects that any outliers might have on regression coefficients [7,9,30,50,53,54,73–76]. Tests of natural log compared to base 10 log regressions also produced the same SVL predictions for large test specimens in this study, so natural log was used in all regressions for consistency. All regression analyses were performed using PAST v.4.03 [72].

To test the assumption that anatomical proportions are conserved between extant and extinct members within any particular lizard lineage represented in this study, the SVL values predicted from individual bones were compared with those measured from complete skeletons of extant taxa from the same lineage and, when available, fossil congeners as well (S1–S7 Tables). The fossil test specimens included three of the few available complete fossil lizard skeletons that were measured: a large-bodied varanid (*Saniwa ensidens*, FMNH PR 2378, SVL 425 mm, which is smaller than the largest extant varanid measured in this study, *V. exanthematicus*, SVL 770 mm; Fig 4A), a small-bodied anguid (cf. *Parophisaurus pawneensis*, YPM VP 060609, SVL 138 mm), and a very small-bodied iguanian (*Afairiguana avius*, FMNH PR 2379, SVL 46 mm; Fig 5B). For each SVL regression tested, the 95% confidence interval for the SVL estimate of each test specimen was calculated by adding or subtracting the standard error of the regression (S1–S7 Tables).

No complete dry skeletonized specimens of extant *Shinisaurus* were available to measure. Only one dry skull and four wet preserved specimens of this monotypic family were found in collections. Fortuitously, however, the holotype and only known complete skeleton of an extinct taxon referred to Shinisauridae, *Bahndwivici ammoskius* (FMNH PR 2260, SVL 154 mm; Fig 5A) [77] had the same head length: SVL ratio as the average calculated for extant *Shinisaurus* based on wet preserved specimens (0.23 vs. 0.22, n = 4; S2 Dataset). Therefore, ratios of individual cranial elements to SVL from the *Bahndwivici* skeleton were used to estimate SVL for the handful of fossil specimens referred to Shinisauridae (S1 Dataset).

## Estimations and new equations for body mass

It has been shown that body mass scales linearly with SVL in extant lizards with limbs [16,25]. This relationship allowed researchers to previously develop allometric equations useful for estimating lizard body mass based on SVL at the family level (S8 Table) [25,42,43] as well as a general SVL-to-mass equation for all lizards (RMA: Body Mass in grams = (((3.088±0.067)*SVL)–(4.852±0.128))/1000, $R^2$ = 0.946) [25]. Body mass for each fossil specimen in the dataset presented here was calculated using the published family-level equations (S8 Table), as well as the published general RMA equation for comparison (S1 Dataset).

This study also generated new equations for estimating lizard body mass directly from individual elements (S1–S7 Tables). This was done by substituting an equation from this study for estimating SVL from a specific element for a given family (e.g., head length-to-SVL for Anguidae, S1 Table) into the published equation for estimating lizard body mass from SVL for that same family (S8 Table) [25,42,43], producing a new equation for calculating mass directly from that specific element (e.g., head length-to-mass for Anguidae, S1 Table). Thus, when applied to the fossil lizard specimens in S1 Dataset, the new element-to-mass equations produced the same body mass estimates as those obtained from the previously published equations (S1 Dataset and S8 Table).

## Analysis of body size evolution

Maximum and mean SVL and mass were used as proxies for understanding patterns of lizard body size evolution through the Paleogene in the Western Interior of North America. To generate these data, the novel equations produced in this study for estimating body length and the published equations referenced in S8 Table for calculating body mass were applied to all fossil lizard specimens in S1 Dataset. When estimating body length or mass for a fossil specimen that included more than one complete cranial or limb element, the corresponding regression that offered the lowest standard error of the estimate (SEE) was used (S1–S7 Tables). All analyses were conducted at the level of crown-group assignments [64–66]. Fossil data were grouped by North American Land Mammal Ages (NALMAs), which compose a relative chronology used to divide the Paleogene based on taxonomic turnover in the mammal fossil record [78,79].

Rarefaction was used to analyze sample sizes though the Paleogene in this study system. S1 Dataset was subsampled by NALMA interval to determine the expected size range per NALMA based on sample size. Each fossil specimen in S1 Dataset was assigned to one of 11 size groups for SVL, each group representing a bin of 100 mm (e.g., 1–99 mm, 100–199 mm, 200–299 mm, etc.; the last group was SVL ≥ 1 m). Individual rarefaction analysis was performed using PAST v4.03 [72].

## Testing for correlation between maximum body size and temperature

RMA linear regression was applied to test the hypothesis of global climatic influence on lizard body size. Global temperatures were regressed to maximum and mean lizard SVL and mass

for each NALMA interval (S9 Table). Temperature data were obtained from Zachos et al. [44,45], a widely used dataset of Cenozoic paleotemperatures derived from marine oxygen isotopes (e.g., [10,50]) and were taken from the NALMA midpoint in ˚C. All data were transformed using natural log. Correlation tests were performed using PAST v4.03 [72].

## Results

### Regression equations for lizard body size

Results investigating the relationships between different skeletal elements and body size across the lizard crown groups represented in this study indicated that the most consistently reliable cranial measurements for estimating SVL were head length (SEE ≤ 0.0997, Anguidae and Iguania) and maxilla length (SEE ≤ 0.133, Anguidae, Varanidae, Scincidae, Xantusiidae, Iguania; Fig 6 and Table 2 and S1–S7 Tables). Regressions based on limb measurements (femur, humerus, or tibia length) also had strong predictive power for Anguidae and Varanidae, the only two groups with fossil limb material available in the fossil dataset (SEE ≤ 0.0815; S1 and S2 Tables). Greater variation in rankings was observed among lizard groups for regressions based on dentary, frontal, and parietal length (SEE 0.0530–0.1571). General equations based on the entire extant dataset proved unsuccessful because none passed tests for both homoskedasticity and normal distribution of residuals (S10 Table).

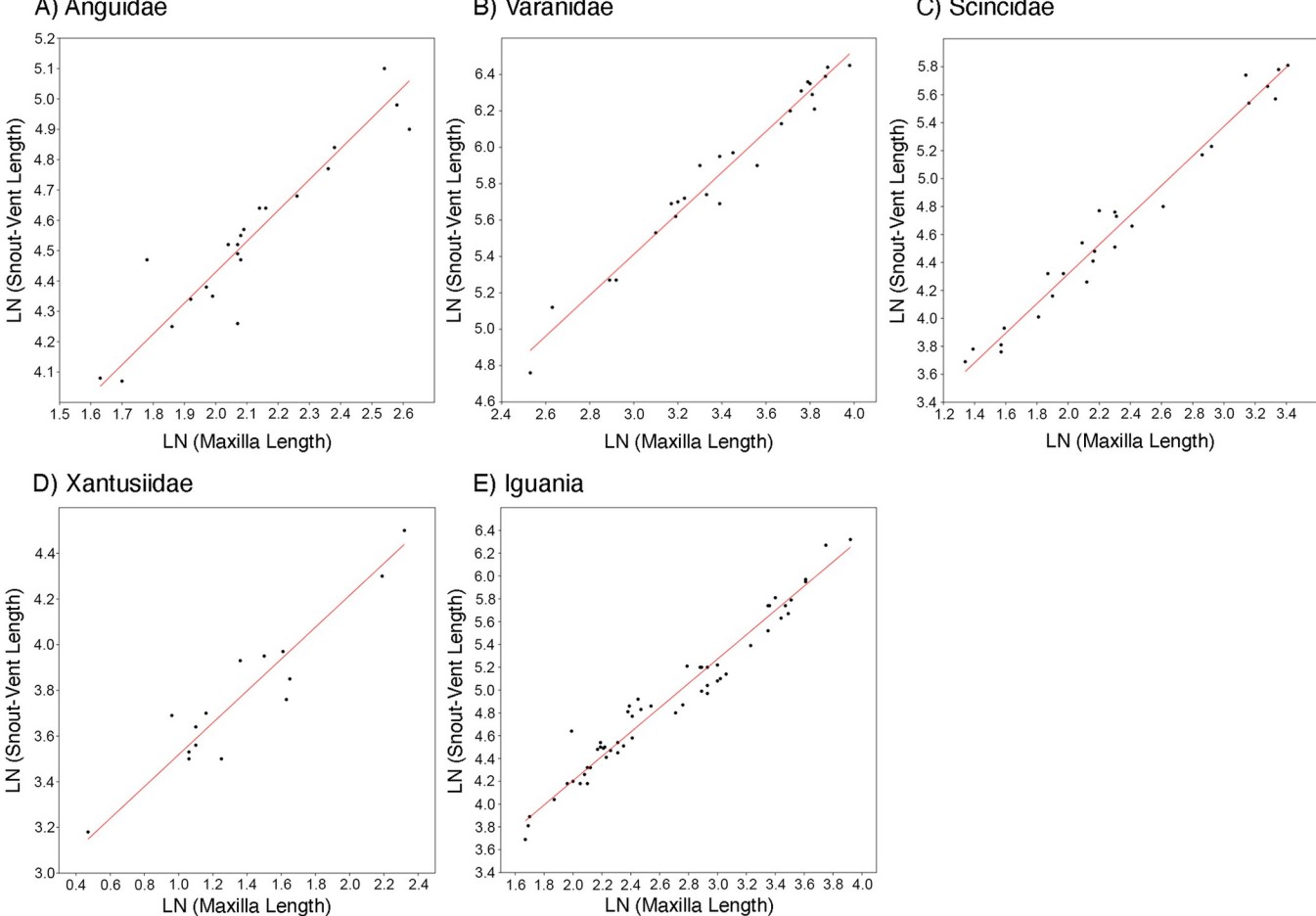

**Fig 6. Regressions of maxilla length to snout-vent length for all extant lizard groups sampled.** Only the groups for which a maxilla length regression was generated are included. See Table 2 for regression equations and statistics. All regressions are transformed using the natural log (LN).

**Table 2. Regression functions and associated statistics for snout-vent length estimation from individual bones.** All functions were transformed using the natural log (LN). Standard error for slope and intercept terms are listed after the slope and intercept in the equations. Functions are listed in order of increasing standard error of the estimate (SEE) for each group. If a particular fossil specimen included more than one complete cranial or limb measurement, the applicable regression that offered the lowest SEE was used. P(uncorrelated) < 0.001 for all functions.

| Group | Predictor | n | Slope b | Standard error of b | Intercept | Standard error of intercept | $R^2$ | SEE |
|---|---|---|---|---|---|---|---|---|
| Anguidae | Skull Length | 46 | 0.946 | 0.0417 | 1.71 | 0.129 | 0.91 | 0.0740 |
| | Femur Length | 43 | 0.888 | 0.0410 | 2.46 | 0.101 | 0.91 | 0.0742 |
| | Mandible Length | 43 | 0.921 | 0.0424 | 1.82 | 0.129 | 0.91 | 0.0760 |
| | Humerus Length | 41 | 0.889 | 0.0451 | 2.57 | 0.106 | 0.90 | 0.0815 |
| | Parietal Length | 24 | 1.03 | 0.0658 | 2.29 | 0.148 | 0.91 | 0.0853 |
| | Maxilla Length | 22 | 1.02 | 0.0803 | 2.40 | 0.170 | 0.88 | 0.0992 |
| | Frontal Length | 22 | 1.27 | 0.111 | 2.13 | 0.216 | 0.85 | 0.120 |
| | Dentary Length | 24 | 0.898 | 0.0816 | 2.43 | 0.195 | 0.82 | 0.121 |
| Varanidae | Tibia Length | 27 | 0.953 | 0.0282 | 2.43 | 0.103 | 0.98 | 0.0695 |
| | Maxilla Length | 25 | 1.13 | 0.0423 | 2.04 | 0.145 | 0.97 | 0.0837 |
| | Dentary Length | 23 | 1.13 | 0.0845 | 1.86 | 0.305 | 0.88 | 0.157 |
| Xenosauridae | Dentary Length | 12 | 0.36 | 0.164 | 0.921 | 0.450 | 0.85 | 0.0530 |
| | Parietal Length | 12 | 1.03 | 0.154 | 2.17 | 0.369 | 0.78 | 0.0661 |
| | Frontal Length | 12 | 1.05 | 0.185 | 2.16 | 0.435 | 0.69 | 0.0787 |
| Teiidae | Dentary Length | 26 | 0.677 | 0.0686 | 2.70 | 0.165 | 0.75 | 0.0995 |
| Scincidae | Maxilla Length | 27 | 1.06 | 0.0352 | 2.20 | 0.0850 | 0.97 | 0.115 |
| | Dentary Length | 27 | 1.12 | 0.0426 | 1.74 | 0.115 | 0.96 | 0.133 |
| Xantusiidae | Maxilla Length | 20 | 0.698 | 0.0659 | 2.82 | 0.0944 | 0.88 | 0.119 |
| | Dentary Length | 20 | 0.787 | 0.0869 | 2.58 | 0.134 | 0.89 | 0.124 |
| Iguania | Skull Length | 53 | 1.15 | 0.0248 | 1.06 | 0.0838 | 0.98 | 0.0997 |
| | Maxilla Length | 55 | 1.07 | 0.0303 | 2.07 | 0.0824 | 0.96 | 0.133 |
| | Dentary Length | 55 | 1.22 | 0.0402 | 1.38 | 0.118 | 0.94 | 0.155 |

The group-level regressions generated for body length in this study predicted actual SVL with comparable accuracy between the extant and fossil test specimens (S1–S7 Tables). The actual SVL for each extant and fossil test specimen fell within or close to the 95% confidence interval for the estimated SVL. The percentage difference between predicted and actual SVL of test specimens ranged from 1% (varanid tibia length vs. SVL, SEE = 0.0695; S2 Table) to 36% (anguid frontal length vs. SVL, SEE = 0.120; S1 Table).

When large-bodied outliers were used to test the predictive power of the SVL regressions, the resulting percentage differences between actual and estimated SVL were comparable (see Varanidae, S2 Table) or lower (see Anguidae, S1 Table) than the other test specimens, indicating that those regressions can reliably estimate SVL for large individuals in those families. A large specimen of *Heloderma horridum* (Helodermatidae, e.g., Fig 3E), the skull of which greatly resembles those of the largest anguid fossils in this dataset (Fig 3I), was used as an additional test specimen for the anguid regressions (S1 Table). Helodermatidae is placed close to Anguidae within the group Anguimorpha in both morphological and molecular phylogenetic analyses [68–71], so it is reasonable to consider Helodermatidae as an extant morphological and close phylogenetic analogue to large extinct anguid lizards. The anguid regressions proved comparably accurate in predicting the SVL of the largest helodermatid test specimen compared to the smaller extant and extinct anguid test specimens (S1 Table).

The published family-specific SVL-to-mass equations [25,42,43] and the general SVL-to-mass equation [25] produced similar mass estimates for fossil specimens, with a percentage difference of 1% (Teiidae and Xantusiidae) to 16% (Xenosauridae; S1 Dataset). The novel

element-to-mass equations generated in this study (S1–S7 Tables) were derived from the published family-specific SVL-to-mass equations, thus, these two sets of equations produced the same mass estimates for fossil specimens.

## Body size in fossil lizards through the Paleogene

Maximum lizard SVL remained below 300 mm through the early and middle Paleocene (Puercan (66.0–63.8 Ma)–Tiffanian (60.9–56.2 Ma), Fig 7 and Table 3 and S1 Dataset and S2 Fig and S11 Table). This upper SVL limit increased to almost 500 mm around the Paleocene-Eocene transition (Clarkforkian (56.2–54.9 Ma)). By the Wasatchian (54.9–50.5 Ma), maximum SVL reached 750 mm, and surpassed 1 m by the Bridgerian (50.5–46.2 Ma; Table 3). Lizard body mass followed a similar pattern, with lizards exceeding 1.5 kg (LN(Mass in kg) = 0.405) by the Clarkforkian (Fig 7 and Table 3 and S1 Dataset and S3 Fig and S11 Table).

Both maximum body size and sample size were lower for the middle Eocene: the Uintan, (46.2–39.7 Ma), with n = 4, had a maximum SVL 540 mm and maximum mass 1.74 kg (*Saniwa*

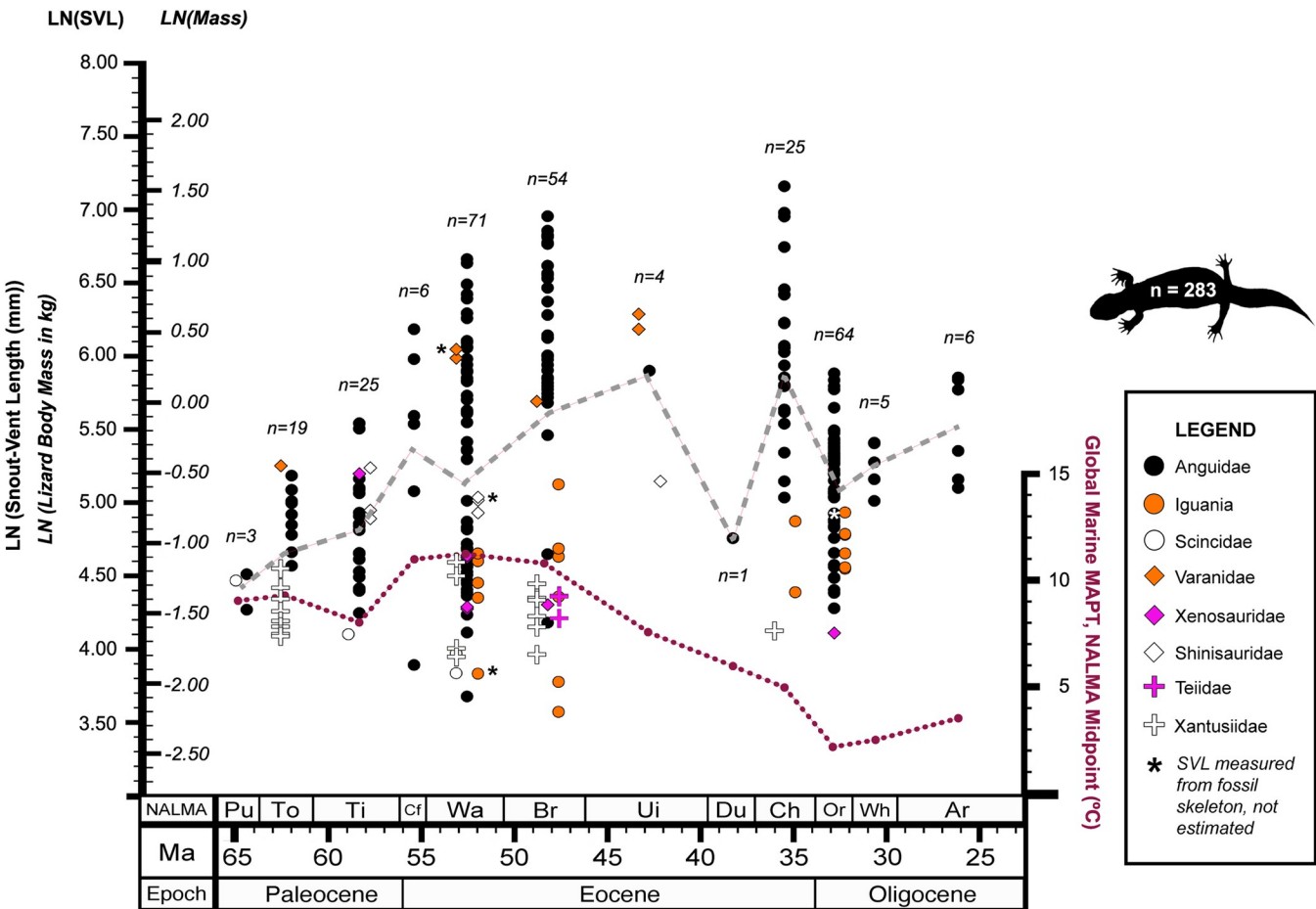

**Fig 7. Body size distribution by taxonomic group for fossil lizards in the Western Interior of North America through the Paleogene.** Body size is plotted as both snout-vent length (SVL) in mm and mass in kg. All measurements were transformed using natural log (LN). Data points marked with an asterisk (*) were measured from complete fossil skeletons. All others were estimated from individual cranial or limb elements using regressions (see S1–S7 Tables). Mass estimates were calculated from SVL using the published equations listed in S8 Table. Dashed line indicates mean LN(SVL) or LN(Mass) per NALMA (see also S2 and S3 Figs). Sample sizes are per North American Land Mammal Age (NALMA). Data points are spread laterally within each NALMA for visibility. Dotted line represents the global mean annual paleotemperature (MAPT in ˚C) for the midpoint of each NALMA interval, as derived from ∂O¹⁸ marine isotope proxies [44,45]. NALMA abbreviations: Pu = Puercan, To = Torrejonian, Ti = Tiffanian, Cf = Clarkforkian, Wa = Wasatchian, Br = Bridgerian, Ui = Uintan, Du = Duchesnean, Ch = Chadronian, Or = Orellan, Wh = Whitneyan, Ar = Arikareean.

**Table 3. Maximum and mean fossil lizard body size estimates per interval.** North American Land Mammal Age (NALMA) ranges are from Barnosky et al. [79]. Mean values are not listed for the Duchesnean because n = 1 for that NALMA. SVL = snout-vent length.

| NALMA | Age Range (Ma) | Crown Group | Binomial | Specimen | Max SVL Estimate (mm) | Mean SVL Estimate (mm) | Max Mass Estimate (kg) | Mean Mass Estimate (kg) |
|---|---|---|---|---|---|---|---|---|
| Puercan | 66.0–63.8 | Anguidae | *cf. Odaxosaurus piger* | UCM 34991 | 91 | 83 | 0.311 | 0.276 |
| Torrejonian | 63.8–60.9 | Varanidae | cf. Varanidae indet. | UCMP 2671 | 191 | 113 | 0.615 | 0.373 |
| Tiffanian | 60.9–56.2 | Anguidae | *Melanosaurus sp.* | UCM 98615 | 256 | 133 | 0.885 | 0.451 |
| Clarkforkian | 56.2–54.9 | Anguidae | *Melanosaurus maximus* | UMMP 74618 | 487 | 269 | 1.69 | 0.932 |
| Wasatchian | 54.9–50.5 | Anguidae | *Melanosaurus sp.* | UCMP 154536 | 788 | 232 | 2.74 | 0.793 |
| Bridgerian | 50.5–46.2 | Anguidae | *Glyptosaurus sylvestris* | USNM 12590 | 1056 | 408 | 3.67 | 1.41 |
| Uintan | 46.2–39.7 | Varanidae | *Saniwa ensidens* | FMNH UC 1719 | 540 | 391 | 1.74 | 1.28 |
| Duchesnean | 39.7–37.0 | Anguidae | Anguidae indet. | CM 42469 | 116 | N/A | 0.399 | N/A |
| Chadronian | 37.0–33.9 | Anguidae | *Helodermoides sp.* | UW 11057 | 1296 | 459 | 4.51 | 1.59 |
| Orellan | 33.9–31.8 | Anguidae | *Peltosaurus sp.* | UCM 20877 | 360 | 176 | 1.25 | 0.604 |
| Whitneyan | 31.8–29.5 | Anguidae | *Peltosaurus sp.* | SMM P81.8.71 | 224 | 194 | 0.775 | 0.669 |
| Arikareean | 29.5–23.0 | Anguidae | *Peltosaurus sp.* | UNSM 81001 | 351 | 262 | 1.22 | 0.906 |

*ensidens*, Table 3), about half that of the maximum size for the previous interval. Only one small specimen was recovered from the Duchesnean (39.7–37.0 Ma, 116 mm, 0.399 kg, Anguidae indet., Table 3). Sample sizes were higher just before and after middle Eocene (early Eocene: Wasatchian, n = 71, and Bridgerian, n = 54; late Eocene: Orellan, 33.9–31.8 Ma, n = 25). Rarefaction analysis (S4 Fig) demonstrated that the number of expected size groups recovered per NALMA from this fossil dataset peaked and leveled off around n = 50.

The late Eocene record (Chadronian, 37.0–33.9 Ma) featured several large individuals, including the largest body size estimated in this study (1.3 m, 4.5 kg, *Helodermoides sp*., Table 3). However, by the early Oligocene (Orellan), maximum body size dropped considerably (360 mm, 1.25 kg, *Peltosaurus sp.*) and remained below that limit through the late Paleogene.

Global temperature did not predict maximum or mean lizard body size per NALMA. RMA linear regression tests did not indicate any correlation between global marine temperatures and maximum SVL (LN(Max Lizard SVL) = $((1.30\pm0.431)^*$LN(Global Marine MAPT)) + $(3.57\pm0.855)$, $R^2 = 0.02$, p(uncorr.) = 0.71, SEE = 1.10) or mean SVL (LN(Mean Lizard SVL) = $((-0.910\pm0.303)^*$LN(Global Marine MAPT)) + $(7.06\pm0.606)$, $R^2 = 0.0004$, p(uncorr.) = 0.95, SEE = 0.812) per NALMA (Figs 7 and S5 and S9 Table). Results were similar for tests of global marine temperatures vs. maximum mass (LN(Max Lizard Mass) = $((1.32\pm0.436)^*$LN(Global Marine MAPT)) + $(-2.13\pm0.866)$, $R^2 = 0.02$, p(uncorr.) = 0.74, SEE = 1.12) or mean mass (LN(Mean Lizard Mass) = $((-0.798\pm0.262)^*$LN(Global Marine MAPT)) + $(1.02\pm0.518)$, $R^2 = 0.03$, p(uncorr.) = 0.59, SEE = 0.651).

## Discussion

### Strength of body size regression equations

This study found skull length to be the strongest cranial element predictor of SVL in Anguidae and Iguania, in agreement with previous studies [12,22,23]. Limb bones were also strong

predictors of SVL in Anguidae (femur and humerus length) and Varanidae (tibia length), as was previously demonstrated in crocodylians [28]. These elements were not available for the other lizard crown groups investigated in this study. Among groups, maxilla length provided the most consistently accurate SVL estimate compared to SVL measurements from test specimens (Fig 6 and S1 and S2 Tables and S5–S7 Tables), followed by dentary length (S1–S7 Tables). The body mass equations could not be tested for accuracy in the same way because body mass data were not available for the extant test specimens. In terms of deviation from the general equation for all lizards, the body mass estimations from equations for Anguimorpha (used for Shinisauridae), Iguania, Teiidae, Varanidae, and Xantusiidae were the closest to the general estimation, with a percentage difference of 1–5% (S1 Dataset and S8 Table).

Attempts to derive general equations for calculating SVL from individual bones proved unsuccessful in this study (S10 Table). This might be because cranial element morphology and proportions often vary between closely related lizard groups [80]. Meiri's [25] general equation for predicting mass from SVL for lizards (see S1 Dataset) was likely more viable because the relationship between SVL and mass appears to be more consistent than the relationship between head length or any individual element to SVL across lizard taxonomic groups.

Separate regressions for males and females were not generated because most extant specimens sampled did not have a sex indicated. However, it was not an issue for this study because any fossil lizard dataset must be treated as an averaged sample across sexes. Although female lizards often have shorter SVLs than males [4], there is no reliable way to tell the sex of an individual fossil lizard specimen based on isolated bones.

The same is largely true for ontogenetic stages represented in a fossil dataset. Adult members of an extinct taxon can be reasonably identified if a large sample of referred individuals that reach a consistent maximum size range is present (e.g., large fossil anguids, S1 Dataset) or if some of those specimens include long bones with fused epiphyses. Complete anatomical elements from adult individuals are also more likely to get preserved as fossils. Juveniles are less likely to get preserved as fossils because they are frequently consumed by predators and their bones are more delicate [81]. This preservation bias toward adult specimens would not affect the resulting patterns of maximum body size for any NALMA with a decent sample size, even if some subadult or juvenile specimens were present in the fossil dataset. For these reasons, and to avoid skewing the regression equations with outliers, juvenile specimens were excluded from the extant dataset.

## Body size evolution among Paleogene lizard groups

Anguids are known to be the most abundant lizards in the Paleogene fossil record of the Western Interior [82–84] and that was reflected in their dominance of the fossil dataset presented here (S1 Dataset). Anguids also accounted for the largest individuals in the dataset and most of the maximum sizes in each NALMA, with the exception of a few varanids (Fig 7 and S2 and S3 Figs and Table 3 and S11 Table). Among the eight lizard crown groups sampled, only those two reached SVLs over 200 mm and masses over 1 kg. Interestingly, the largest extant limbed anguid today, *Diploglossus millepunctatus* (maximum SVL 280 mm), does not approach the size of the largest North American anguids from the Eocene, nor do the largest lizards occurring in North America today (*Heloderma horridum*, maximum SVL 520 mm [85], similar to Fig 3E; *Iguana iguana*, maximum SVL 580 mm [4]). Conversely, Varanidae includes both the largest extant lizard (*Varanus komodoensis*, which exceeds the adult size of the largest Eocene anguid at around 1.5 m SVL [4,86]) and the largest known terrestrial lizard in the fossil record (*Varanus priscus*, known from Pleistocene deposits in Australia, with SVL at least 3 m [87,88]), but varanids are now restricted to the Eastern Hemisphere [13].

The other lizard groups in this geotemporal system (Iguania, Scincidae, Shinisauridae, Teiidae, Xantusiidae, Xenosauridae) did not experience an increase in body size range (Fig 7 and S11 Table). Small-bodied lizards dominated the North American Paleogene lizard record. The maximum and mean body sizes for most lizard groups did not exceed 200 mm in SVL (LN (SVL) = 5.30, S2 Fig and S11 Table) or 1 kg in body mass (LN(Mass) = 0.00, S3 Fig and S11 Table).

The observed maximum body size for the middle Eocene (540 mm, 1.74 kg, *Saniwa ensidens* (Varanidae), Figs 4A and 7 and Table 3 and S11 Table) was much lower than that of the early or late Eocene. Fewer outcrops are available from this interval in the U.S. Western Interior, so this result was likely an artifact of poor sampling (Fig 7 and Table 3). Rarefaction analysis (S4 Fig) supported the assumption that the NALMAs with small sample sizes in this study did not capture the full range of lizard body size that was likely present during those intervals.

The decrease in maximum body size in the early Oligocene cannot be attributed to low sample size because the Orellan (33.9–31.8 Ma) was very well sampled (n = 64) compared to other NALMAs (Figs 7 and S4). In fact, sample size did not correlate strongly with maximum estimated SVL across the entire dataset ($R^2$ = 0.23, p(uncorr.) = 0.14 for sample size *n* vs. LN (Max SVL)). The observed late Paleogene reductions in maximum body size likely represent the local or complete extinction of the largest Paleogene lizard taxa in the region [89,90]. Even in extant lizards, extinction risk increases with body size [3]. Among the eight lizard groups documented in this study from the Paleogene record of this region, only Anguidae, Scincidae, Teiidae, and Xantusiidae still occur in the area. Xenosaurids and the iguanians included in this study that occurred in the Western Interior during the Paleogene have since become locally extinct in the region; varanids and shinisaurids no longer occur anywhere in North America [4,11,13,86].

In general, the observed patterns may reflect some taphonomic or preservation bias (especially with respect to smaller bones), gaps in the rock record, and selective sampling by fossil collection crews. However, rarefaction analysis (S4 Fig) indicated that this study adequately sampled the upper range of body size diversity represented overall in this fossil record, especially from thoroughly documented areas like the Bighorn and Green River Basins (S1 Dataset; for further discussion, see the censuses of microvertebrate localities in this system by Smith [57–61]). Preservation bias toward complete elements from adult individuals may have mitigated other sources of bias when reconstructing maximum body size through geologic intervals.

## Climate as a potential driver of body size evolution

Environmental temperature can influence body size evolution in reptiles [3,9,32,91–93]. Metabolic rate increases with temperature and scales with body size in all poikilothermic vertebrates [1,94,95], including lizards [16,96]. Changes in environmental temperature are known to correlate globally with changes in maximum body size in extant lizards (exclusive of snakes [15,26]). Increased temperatures around the Paleocene-Eocene Thermal Maximum (PETM, 56 Ma; [45,97]) could have elevated metabolism and released a physiological constraint on maximum size for some lizard taxa. Gigantism has been documented in other fossil reptiles in conjunction with elevated temperatures, including a boid snake (*Titanoboa*, total length 13 m, 60–58 Ma, Colombia [32]), a caimanine crocodyliform (*Purussaurus*, total length 12.5 m, 8 Ma, northern South America [98]), and a varanid lizard (*Varanus priscus*, total length up to 5.7 m, around 50 ka, Australia [87,88]).

Here, fossil lizard body size showed no relationship to global temperature through the Paleogene (S5 Fig). It is possible that the relationship observed today is not detectable across

scales of millions of years, perhaps due to taphonomic or sampling bias in the fossil record. It might also be more pertinent to compare these data with terrestrial paleotemperature proxies from the same geotemporal system sampled in this study. This could be a fruitful area for continued research to build on the data presented here.

Alternatively, body size evolution patterns observed here could reflect indirect effects of climate [99]. Temperatures were warmer during both the summer and winter months around the PETM, and most precipitation came during the summer [100–102]. This would have allowed lizards to be active and growing during a longer portion of the year [99,103] and could also have led to greater food availability from increased productivity, especially in the summer months [104–106]. Previous studies found elevated growth rates in extant tropical lizards occupying areas that received greater precipitation [107]. Attaining large body size in a closed tropical forest system with an equable climate, like that present in the Western Interior in the early Eocene [104,105,108,109], would also have allowed lizards to spend less time and energy on active thermoregulation [1,110]. Large body size sometimes corresponds with thermoconforming behavior and nocturnal activity in extant lizards [3,12,103], and these strategies could have reduced competition with diurnal organisms. Herbivory can also lead to large body size in lizards [3,4,96]. This has been inferred for at least one giant fossil lizard [9]. However, the dentition of the largest lizards in this study were more indicative of omnivorous or insectivorous (Anguidae, Fig 3A) and carnivorous diets (Varanidae, Fig 4B; see [22,111]), rather than herbivory.

Lizards can maintain smaller body sizes in warm climates as a result of behavioral thermoregulation [110], competition, predation, or specializations related to resource zones of small taxa [93,112]. This might explain why not all lizard groups in this fossil dataset exhibited an increase in body size. These results may reflect an ecological hierarchy in the Paleogene communities of the Western Interior similar to extant ecosystems in which large-bodied lizards are less abundant than small-bodied lizards [3].

## Conclusions

This study generated novel equations for estimating fossil lizard body size from isolated anatomical elements, with skull length, limb bone length, and maxilla length providing the most accurate estimates for SVL. Body mass can also be calculated from individual bones based on these equations. Application of these equations to the Paleogene record of lizards in the Western Interior of North America offers the first survey and reconstruction of body size across crown lizard groups through a prolonged geologic interval and across a large continental interior region. The findings indicate that lizard body size peaked in the early Eocene in this system. Only two lizard groups, Anguidae and Varanidae, reached large body sizes exceeding 0.5 m in SVL and 1.5 kg in mass. Body size range did not change considerably for other groups. Maximum lizard body size decreased across the Western Interior in the early Oligocene. The methods presented here can be applied to other lizard clades to generate reasonably accurate body size estimates for fossil taxa and to study patterns of body size evolution across higher taxonomic groups.

## Supporting information

**S1 Fig. Protocol for estimating snout-vent length of fossil lizards.**
(TIF)

**S2 Fig. Mean and maximum body length in fossil lizards by taxonomic group per NALMA.**
Body length is measured as snout-vent length (SVL) in mm. All measurements were

transformed using natural log (LN). Dotted line = maximum. Dashed line = mean. Lines do not connect with the single Duchesnean datapoint since it likely does not accurately represent a maximum or mean value for that NALMA. North American Land Mammal Age (NALMA) abbreviations: Pu = Puercan, To = Torrejonian, Ti = Tiffanian, Cf = Clarkforkian, Wa = Wasatchian, Br = Bridgerian, Ui = Uintan, Du = Duchesnean, Ch = Chadronian, Or = Orellan, Wh = Whitneyan, Ar = Arikareean.
(TIF)

**S3 Fig. Mean and maximum body mass in fossil lizards by taxonomic group per NALMA.** Mass estimates were calculated from snout-vent length (SVL) using the published equations listed in S8 Table. All measurements (in kg) were transformed using natural log (LN). Dotted line = maximum. Dashed line = mean. Solid black line indicates 1 kg threshold (LN(1) = 0.00). Lines do not connect with the single Duchesnean datapoint since it likely does not accurately represent a maximum or mean value for that NALMA. North American Land Mammal Age (NALMA) abbreviations: Pu = Puercan, To = Torrejonian, Ti = Tiffanian, Cf = Clarkforkian, Wa = Wasatchian, Br = Bridgerian, Ui = Uintan, Du = Duchesnean, Ch = Chadronian, Or = Orellan, Wh = Whitneyan, Ar = Arikareean.
(TIF)

**S4 Fig. Rarefaction curve showing the number of expected size groups for NALMA sample sizes within the fossil lizard dataset.** To conduct this analysis, each fossil specimen in S1 Dataset was assigned to one of 11 size groups, each group representing a bin of 100 mm (e.g., 1–99 mm, 100–199 mm, 200–299 mm, etc.; the last group was ≥ 1 m). S1 Dataset was subsampled here by North American Land Mammal Age (NALMA) interval. The y-axis in this graph shows the total number of size groups recovered in the sample for each NALMA interval. NALMA abbreviations: Pu = Puercan, To = Torrejonian, Ti = Tiffanian, Cf = Clarkforkian, Wa = Wasatchian, Br = Bridgerian, Ui = Uintan, Du = Duchesnean, Ch = Chadronian, Or = Orellan, Wh = Whitneyan, Ar = Arikareean. Individual rarefaction analysis was performed using PAST v4.03 [72].
(TIF)

**S5 Fig. Global temperature vs. lizard body size by NALMA. A)** Global marine mean annual paleotemperature (MAPT) vs. maximum lizard SVL as a proxy of maximum body size. Function: LN(Max Lizard SVL) = ((1.30±0.431)*LN(Global Marine MAPT)) + (3.57±0.855), $R^2$ = 0.02, p(uncorr.) = 0.71, SEE = 1.10. **B)** Global marine MAPT vs. mean lizard SVL. LN(Mean Lizard SVL) = ((-0.910±0.303)*LN(Global Marine MAPT)) + (7.06±0.606), $R^2$ = 0.0004, p (uncorr.) = 0.95, SEE = 0.812. The graphs for correlations of global temperature with maximum and mean mass had the same spread of data. Temperature data are from Zachos et al. [45] and were taken from the NALMA midpoint in ˚C. All data were transformed using natural log. Regression analysis was performed using PAST v4.03 [72].
(TIF)

**S1 Table. Anguidae regression equations and tests.** Sample sizes indicate number of extant anguid specimens that included the element used in the given regression. Regression equations are ranked in order of increasing standard error of the estimate (SEE). Minimum and maximum SVL estimates for test specimens are based on the standard errors for the slope and intercept of the given regression. Equations for body mass from individual bones are based on the relevant equation listed in S8 Table.
(XLSX)

**S2 Table. Varanidae regression equations and tests.** Sample sizes indicate number of extant varanid specimens that included the element used in the given regression. Regression equations are ranked in order of increasing standard error of the estimate (SEE). Minimum and maximum SVL estimates for test specimens are based on the standard errors for the slope and intercept of the given regression. Equations for body mass from individual bones are based on the relevant equation listed in S8 Table.
(XLSX)

**S3 Table. Xenosauridae regression equations and tests.** Sample sizes indicate number of extant xenosaurid specimens that included the element used in the given regression. Regression equations are ranked in order of increasing standard error of the estimate (SEE). Minimum and maximum SVL estimates for test specimens are based on the standard errors for the slope and intercept of the given regression. Equations for body mass from individual bones are based on the relevant equation listed in S8 Table.
(XLSX)

**S4 Table. Teiidae regression equations and tests.** Sample sizes indicate number of extant teiid specimens that included the element used in the given regression. Regression equations are ranked in order of increasing standard error of the estimate (SEE). Minimum and maximum SVL estimates for test specimens are based on the standard errors for the slope and intercept of the given regression. Equations for body mass from individual bones are based on the relevant equation listed in S8 Table.
(XLSX)

**S5 Table. Scincidae regression equations and tests.** Sample sizes indicate number of extant scincid specimens that included the element used in the given regression. Regression equations are ranked in order of increasing standard error of the estimate (SEE). Minimum and maximum SVL estimates for test specimens are based on the standard errors for the slope and intercept of the given regression. Equations for body mass from individual bones are based on the relevant equation listed in S8 Table.
(XLSX)

**S6 Table. Xantusiidae regression equations and tests.** Sample sizes indicate number of extant xantusiid specimens that included the element used in the given regression. Regression equations are ranked in order of increasing standard error of the estimate (SEE). Minimum and maximum SVL estimates for test specimens are based on the standard errors for the slope and intercept of the given regression. Equations for body mass from individual bones are based on the relevant equation listed in S8 Table.
(XLSX)

**S7 Table. Iguania regression equations and tests.** Sample sizes indicate number of extant iguanian specimens that included the element used in the given regression. Regression equations are ranked in order of increasing standard error of the estimate (SEE). Minimum and maximum SVL estimates for test specimens are based on the standard errors for the slope and intercept of the given regression. Equations for body mass from individual bones are based on the relevant equation listed in S8 Table.
(XLSX)

**S8 Table. Published equations for estimating lizard body mass from snout-vent length.** These equations were used to estimate mass for fossil lizards as listed in S1 Dataset and to generate equations for estimating body mass from individual bones (S1–S7 Tables).
(XLSX)

**S9 Table. Data for correlating SVL with temperature.** Temperature data came from Zachos et al. [45] and were taken from the NALMA midpoint in ˚C. The Duchesnean was not included because the maximum SVL estimate for that NALMA was exceptionally low and based on a single data point.
(XLSX)

**S10 Table. Tests of general regression equations.** Sample sizes indicate number of extant specimens that included the element used in the given regression. Regression equations are ranked in order of increasing standard error of the estimate (SEE). None of these equations passed tests for both homoskedasticity and normal distribution of residuals, thus, none of these equations were used in this study.
(XLSX)

**S11 Table. Mean and maximum fossil lizard snout-vent length and body mass by taxonomic group per NALMA.** These natural log-transformed data were used to generate the graphs in S2 and S3 Figs.
(XLSX)

**S1 Dataset. Fossil lizard data.** Measurements, locality information, and body size estimates for all fossil lizard specimens sampled. Family-specific mass was calculated from equations listed in S8 Table [25,42,43]. General mass estimate (across all lizard groups) was calculated using the embedded equation from Meiri [25].
(XLSX)

**S2 Dataset. Extant lizard data.** Data for all extant lizard specimens measured.
(XLSX)

**S1 File. Supporting information references.** Additional references cited only in Supporting Information.
(DOCX)

## Acknowledgments

The author wishes to thank K. Padian, J. McGuire, A. Barnosky, and I. Wang for their mentorship and helpful comments on this work, as well as J. Head, who mentored the research that started this project. The following individuals provided assistance and access to museum collections: M. Norell, C. Mehling, J. Meng, D. Kizirian, L. Vonnahme, M. Arnold (AMNH); J. Vindum, L. Scheinberg, E. Ely (CAS); M. Lamanna, J. Padial, A. Henrici, S. Rogers (CM); T. Lyson, L. Ivy, K. MacKenzie (DMNH); J. Bloch, R. Hulbert, J. Bourque (FLMNH); P. Makovicky, L. Grande, A. Resetar, W. Simpson, A. Stroupe, J. Mata (FMNH); A. Aase (FOBU); N. Smith, S. McLeod, V. Rhue (LACM); J.Rosado (MCZ); J. McGuire, C. Spencer (MVZ); T. Williamson (NMMNH); A. Farke, G. Santos (RAM); A. Hastings, E. Whiting (SMM); C. Bell, M. Brown (TMM); T. Culver, N. Ridgwell (UCM); P. Holroyd, C. Marshall (UCMP); J. Wilson, A. Rountrey (UMMP); R. Hunt, Jr., P. Freeman, G. Corner, T. Labedz (UNSM); H. Dieter-Sues, M. Carrano, M. Brett-Surman, A. Millhouse (USNM); L. Vietti (UW); C. Sidor, R. Eng, M. Rivin (UWBM); J. Gauthier, C. Norris, D. Brinkman (YPM). The author thanks the following for helpful discussions: C. Badgley, C. Bell, B-A. Bhullar, J. Bloch, W. Clemens[†], C. Crumly, J. Gauthier, W. Gearty, P. Gingerich, L. Grande, A. Gunderson, P. Holroyd, E. Holt, R. Huey, R. Hunt, Jr., M. Kemp, D. Meyer, D. Miles, J. McGuire, J. Parham, H. Petermann, R. Secord, M. Stocker, R. Sullivan, D. Watkins, L. Weaver, S. Werning, E. Whiting, C. Williams, the MVZ

Herp Group, the entire UCMP community. C. Bell also provided access to obscure literature. D. Meyer provided measurements of an unpublished specimen. G. Brown (UNSM) prepared a new *Peltosaurus* specimen (UNSM 12102) containing critical limb material. F. Vasconcellos provided the photo in Fig 3E. B. Chelemedos and C. Wong assisted with data collection. E. Whiting provided instruction on using QGIS to make maps. J. Lipps helped to design a visually accessible color palette for Fig 2. The author thanks C. Meloro, S. Scarpetta, and two anonymous reviewers for their constructive comments which greatly helped to improve the manuscript.

## Author Contributions

**Conceptualization:** Sara J. ElShafie.

**Data curation:** Sara J. ElShafie.

**Formal analysis:** Sara J. ElShafie.

**Funding acquisition:** Sara J. ElShafie.

**Investigation:** Sara J. ElShafie.

**Methodology:** Sara J. ElShafie.

**Project administration:** Sara J. ElShafie.

**Resources:** Sara J. ElShafie.

**Software:** Sara J. ElShafie.

**Supervision:** Sara J. ElShafie.

**Validation:** Sara J. ElShafie.

**Visualization:** Sara J. ElShafie.

**Writing – original draft:** Sara J. ElShafie.

**Writing – review & editing:** Sara J. ElShafie.

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
