## [Decision Letter · Decision Letter 0]

11 Mar 2022

PONE-D-22-04244Body size estimation from isolated fossil bones reveals deep time evolutionary trends for lizardsPLOS ONE

Dear Dr. ElShafie,

Thank you for submitting your manuscript to PLOS ONE. After careful consideration, we feel that it has merit but does not fully meet PLOS ONE’s publication criteria as it currently stands. Therefore, we invite you to submit a revised version of the manuscript that addresses the points raised during the review process.

The paper must be re-structured considerably with clearer sections, scopes and aims. It is too long and a lot of details must be moved in the supplementary material. The section about reconstructing body mass must be analysed and discussed more in details providing better statistics to identify best predictive equations and level of errors in predicting fossil SVL / Body mass. The section about body size changes through time must equally be revised and analysed using more updated methods to assess patterns of body size changes through time. The style of the paper also must be made more impersonal and more in line with that of a scientific journal publication. The data presented are definitely of good quality but they need to be presented in a more structured way. Follow careful also minor advises from all the reviewers. 

We look forward to receiving your revised manuscript.

Kind regards,

Carlo Meloro

Academic Editor

PLOS ONE

Journal Requirements:

2. In your manuscript, please provide additional information regarding the specimens used in your study. Ensure that you have reported specimen numbers and complete repository information, including museum name and geographic location. 

For more information on PLOS ONE's requirements for paleontology and archaeology research, see https://journals.plos.org/plosone/s/submission-guidelines#loc-paleontology-and-archaeology-research.

Additional Editor Comments:

This paper covers an interesting topic and provides novel advancement in our understanding of fossil lizard diversity. The main problem is that it is too long and set out as “dissertation-like” style. You should write more in an impersonal style and try avoiding provide too many methodological details in the main text. Most of it [e.g. stats to test regression] are unnecessary details concerning standard stats recommendation all scientists are familiar with. Other methodological choices about restriction of sample size can go in a Supplementary file.

Also the aims and scope of the paper is not entirely clear so you need to streamline your research. There are two main topics you cover: 1. The reconstruction of body mass in lizards. For this I recommend you to follow style similar to what previous authors did on other groups [e.g. see the work of Egi, N. (2001). Body mass estimates in extinct mammals from limb bone dimensions: the case of North American hyaenodontids. Palaeontology, 44(3), 497-528; and Van Valkenburgh 1990. Skeletal and dental predictors of body mass in carnivores. 181±206. In DAMUTH, J. and MACFADDEN, B. J. (eds). Body size in mammalian paleobiology: estimation and biological implications. Cambridge University Press, New York, 397 pp.]. Based on this you need to restructure your result section presenting your equation / slopes and SEE for different equation [one general including all the lizards extant groups measured, and some more detailed by family…juveniles should be excluded in all cases since you are working on skeletons and you should clearly report the SEE, slope and intercept for each subgroup. Log transformation should also be applied in all cases…note generally that log should be on base 10..you need to justify why you are using natural log]. Aim of this section will be to identify the level of error you got for each bone element and what best bone element can be used for predicting SVL and Body mass. Extensive comparison with the work of Meiri 2010 should also be done and the discussion should reflect this.

2. Patterns of body size diversification in North American fossil lizard. This is a second topic that deserve better analytical approach, data presentation and analyses. For this I refer you to the work on mammals again: Alroy, J. (1998). Cope's rule and the dynamics of body mass evolution in North American fossil mammals. Science, 280(5364), 731-734; Alroy, J. (2000). New methods for quantifying macroevolutionary patterns and processes. Paleobiology, 26(4), 707-733. Aim is to explore patterns of body size diversification through time; Huang, S., Eronen, J. T., Janis, C. M., Saarinen, J. J., Silvestro, D., & Fritz, S. A. (2017). Mammal body size evolution in North America and Europe over 20 Myr: similar trends generated by different processes. Proceedings of the Royal Society B: Biological Sciences, 284(1849), 20162361; Lovegrove, B. G., & Mowoe, M. O. (2013). The evolution of mammal body sizes: responses to C enozoic climate change in N orth A merican mammals. Journal of Evolutionary Biology, 26(6), 1317-1329; Saarinen, J. J., Boyer, A. G., Brown, J. H., Costa, D. P., Ernest, S. M., Evans, A. R., ... & Smith, F. A. (2014). Patterns of maximum body size evolution in Cenozoic land mammals: eco-evolutionary processes and abiotic forcing. Proceedings of the Royal Society B: Biological Sciences, 281(1784), 20132049.

These are some examples showing you what can be done to analyse patterns of lizard body size evolution in time. I am not familiar with reptile literature but there are probably similar work that can be followed up. If you just want to analyse the trend in time that would be fine but you provide too much discussion on factors you do not analyse such as ecological / dietary adaptations that although fundamentals to understand your pattern, have not been treated statistically neither quantified. So, in your case it will be better to look at body size patterns through time and in relation to climate [using the Oxygen isotope record] or eventually average size patterns through time and how they change in different families [a simple correlation analysis can help you to test if when one group is evolving large sizes, the other is evolving smaller size [as expected due to competition] or if size changes applies in similar direction in different groups since they all respond the same way to climate [e.g. in the Palaeoecene-Eocene we do expect all lizard to be significantly larger than in other periods]. I understand that your work is not covering all the taxonomic richness but you implemented rarefaction and you can limit your observation to what you sampled.

Below more detailed observations:

I wonder how SVL is identified from the skeleton. Can you find the approximate position of the cloaca in the pelvis region? You should provide a figure of what you measured.

In Fig. 1 you need to show the exact location of fossil localities and perhaps provide a list of localities and dates which I presume can be based on palaeodb or specific publications.

The introduction is fine but some sections are a bit repetitive so try to cut them. It is reported multiple times that there are no data linking specific bones with lizards. Mention this issue only once.

Line 98-99: how many families are you analyzing among extant and fossil species? Are you testing that regression slopes will not change between families? Make this question a bit clearer. Are the fossil crown lizard completely extinct taxonomic groups or can they be linked to extant families in some way?

In Table 1 you can avoid putting Ln and just mention in the caption that all measurements were natural log transformed. If for some you did not do it, you need to explain why. It will make more sense to generate another table. You need to arrange this info in a better way so that for each taxonomic group we can understand how many anatomical elements were regressed [e.g. each anatomical element can be a separate column]

Line 178-202 looks more of supporting information not needed at this stage. Try to make condense the crucial points in a small sentence and transfer these details in the Supplementary

Line 209-213: If you included both skeletonized and wet SVL estimates in the same dataset did you make sure that they correspond and correlates well in specimens where you can verify this? Using X-ray of wet-specimens you can measure the SVL based on the skeleton in a subsample [e.g. 10 random individuals] and verify if it correlates well with measurements taken from external body

Line 217-22-: this can be another source of error because depending on the photos you might get biased measurements. Can you assure us that this is not the case? Do you have species where this can be verified (e.g. species averages SVL from wet measured individuals correspond well with those from photos of website?)?

Caption of Fig. 5: “p(uncorr) < 0” is not good…just report p values as p < 0.0001…why it should be corrected? You should report more importantly the slope and confidence intervals

Line 273-297: This part is too long and convoluted. Simplify and I think you should follow simpler procedure such as reporting for each regression equation the SEE as you can find in Egi, N. (2001). Body mass estimates in extinct mammals from limb bone dimensions: the case of North American hyaenodontids. Palaeontology, 44(3), 497-528.

You should also test differences in slope between families using ANCOVA or any other similar procedure to test if families follow a different allometric pattern or not. One way is to merge data for the whole sample and add family in a simple gls model: SVL~ [Bone L] + family + [Bone_L]*family…if the interaction term is non-significant than you can use a general regression for all fossil predictions. Family-specific regressions are generally more precise but to demonstrate this you must at least report the SEE for each family and for all the dataset. For example see also Van Valkenburgh 1990. Skeletal and dental predictors of body mass in carnivores. 181±206. In DAMUTH, J. and MACFADDEN, B. J. (eds). Body size in mammalian paleobiology: estimation and biological implications. Cambridge University Press, New York, 397 pp.

Also you should always use log transformation and stick with it all the time.

Line 305-311: again, this can go in the supplementary. In theory you can also try to use multiple regression if some fossils are particularly well preserved. For an example see Figueirido, B., Pérez-Claros, J. A., Hunt, R. M., & Palmqvist, P. (2011). Body mass estimation in amphicyonid carnivoran mammals: A multiple regression approach from the skull and skeleton. Acta Palaeontologica Polonica, 56(2), 225-246.

Line 312 onward: this is fine although it will be more re-assuring to test if your slope do not differ between sexed individuals in a model: body_size~bone_L+sex+bone_L*sex. Same should be done for juveniles whose data points might alter sensibly the slope of only adult regression. If that is an improvement on one side, on the other it might be an additional source of error so be careful.

Line 327-374: these details are all for the supplementary and they should be considerably shortened for the publication that at the moment is too long. So, try to summarise in few sentences what you did to predict the fossil aguids and why and for details [e.g. outliers, sample size, consideration of specific taxa] amend that to the supplementary

Line 376-447: same problem here. This section is too long and your solution is not entirely correct. What you should od in case extinct species are outside the range of their closest relative is using a more general allometry that is based on all the data you have available for all the lizards. That might be the best solution. Your explanation is reasonable within the limit that body proportions between different families are similar which is not your case [as you say later in the section]. So to overcome this problem it is safer to apply a general allometry to cover extra large specimens.

Line 454-455: “I came across a couple of fossil dentaries ascribed to Acrodonta, but I did not include them in my dataset because I did not have time to collect a representative sample of extant Acrodontan diversity”. This is not a reasonable sentence/justification remove this and reduce your writing. This is too long.

Page 26 Results: because this is not a method paper and was not presented as a paper where you are going to recommend a method I suggest to change this structure and be more coherent with the structure of Mat & Methods following results in consequential way.

Also the entire line 497-520 should go eventually AFTER you report the equations you generate, the SEE and so on…this can be the start of a Discussion section but NOT of Results. Also your statistically tips are not needed here. Testing homoscedasticity and outliers is standard practise and MUST be done if regression is the method to use. This should be familiar to any scientist, so all these recommendations are superfluous. You need to show what equation produced the best SEE and answer what bone element is more appropriate depending on the taxonomic group. R-square will all be large but SEE is more important. Please read the papers I suggested on mammals and follow that kind of structure in reporting the results.

Line 640 – 800: these details are a bit out of context and were not covered by the paper. There are many more issues to be discussed based on the data you produced, so you should focus more on that. You need to re-focus completely your discussion trying to condense points not covered / tested by your dataset.

Line 806-808: this sentence is kind of suggesting us that your study is not that important. It is good that you acknowledge previous work on diversity pattern but I presume your work is a way forward. You need to re-phrase this work and make consideration about why your study is important! Also in the next section you are making a lot of justifications for your choice…this is not necessary. You need to re-calibrate your writing here.

Reviewers' comments:

Reviewer's Responses to Questions

**Comments to the Author**

1. Is the manuscript technically sound, and do the data support the conclusions?

Reviewer #1: Yes

Reviewer #2: Yes

2. Has the statistical analysis been performed appropriately and rigorously? 

Reviewer #1: Yes

Reviewer #2: Yes

3. Have the authors made all data underlying the findings in their manuscript fully available?

Reviewer #1: Yes

Reviewer #2: Yes

4. Is the manuscript presented in an intelligible fashion and written in standard English?

Reviewer #1: Yes

Reviewer #2: Yes

5. Review Comments to the Author

Reviewer #1: Dear author,

I enjoyed reading this paper and I think that it is an excellent contribution to the field, and it is appropriate for this journal. The manuscript is well written and presents useful data and methodology for squamate paleontologists and biologists, which I imagine will also be useful to researchers studying other vertebrate taxa. The results specific to North American fossil lizards are very interesting and will no doubt impact broader thinking on lizard body size evolution during the Paleogene. I have a number of comments, most of which are relatively minor (phrasing, precision or accuracy of certain terms), and a handful of which will require more thought (adding in background for the focal clades in the introduction and caveats to the identifications of a few fossils in the dataset, potentially making a flowchart or workflow figure for “Protocol for estimating fossil lizard body size,” changes to discussion of glyptosaurine diet). Overall though I think that the paper is in great shape and I am looking forward to seeing it published.

The full comments are in an attached file.

Reviewer #2: This paper can be an important cornerstone in Lepidosaur paleontology, and I highly encourage its publication. Several sections should be considerably shortened with many sections moving to supplementary material (such as special considerations, homoskedasticity etc…). These can be referred to in the main text but not elaborated to the same degree. You need to be very clear in the main text as to how many families, species, and genera you analyzed, both extant and extinct. I didn’t quite understand if you considered unidentified species from a genus to be one species, I elaborate below on how this can be dealt with. It is important to use clade specific equation for calculating body mass from SVL from Meiri 2010 as opposed to the general equation from Pough, this will likely alter the results of body mass that you received. In addition, I believe that Meiri 2010has a more up to date general equation for lizards in cases where clade specific equations are not suitable. I would also recommend that you briefly state how you think the fossil record is (or is not) biased in the study region. Would you expect species from the entire body size range to be represented? For example, in studies of reptiles trapped in amber, readers are told to expect juvenile or very small individuals to make up the majority of specimens (svl ~ 30mm). If you don’t expect a bias even better but explain that. In addition, you write that you excluded certain species, I recommend stating clearly if they may stand out as exceptionally large or small and alter your results if they would have been included.

You should explore more how changes in body size can be interpreted in relation to the environment. For example, the Eocene greenhouse potentially allowed for evolution of large size (Titanoboa). Or did you see a radiation into new body size niches after the kpg impact? How long did this take? This felt the most lacking to me.

Figures 6 and 7 are not log-transformed and figure 8 is, they should all be log-transformed. Log10 transformation is much better for graphing purposes, but it doesn’t really matter. You may want to try to show changes in mass between different clades through time, either by connecting a line through their mean/median mass or some other method. Up to you though this isn’t entirely necessary.

These comments are made to be constructive and helpful. As I said, I think that this paper has the potential to be an important contribution for those working in Lepidosaur paleontology but requires restructuring and reanalysis and I therefore recommend major revisions while emphasizing its importance.

Jacob Dembitzer

General comments:

Abstract – should be one paragraph

37 – 41 – I don’t understand why lizards are particularly important group compared to bats or frogs? In any case, I would not recommend stating why lizards are important based on their relationship with other animals (as you write they are a food source for other species and regulate populations). Surely their lives are more important than that. I would recommend that you instead emphasize traits that make THEM special. High population densities in small areas, ability to occupy inhospitable regions, high dispersal ability etc. see pough 1980. You can add that squamates are the most diverse clade of tetrapod on earth and in addition discuss how they relate to other animals.

47 – 49 – this can be part of the previous paragraph

53 – references 15 and 16 pertain to crocodilians which are not very closely related to lizards. I recommend that you either change these references or explain that this is part of the knowledge gap that you aim to fill even though data are available for other clades (crocodilians, dinosaurs, mammals). The second option is probably better.

56 -60 – same as last comment. I think you should show that few data are available for lepidosaurs even though there is a wealth of data for species poor crocodilians and other clades.

89-93 – explain why the Paleogene is an interesting period (after mass extinction event, explosive radiation of squamates, etc…). explain why you want to study this period and it’s significance other than simply stating that the data are readily available.

134-140 – did you collect data from the literature or museums? Please state this. Also you should state how many species of fossil and extant lizards you sampled and not simply specimens.

178 – as I wrote above, please state how many extant species

190-191 – it’s still a bit unclear how you actually got the data. Did you measure the specimens yourself in museums? With digital calipers? Please elaborate for both extant and extinct species.

194 – I recommend you site wooley et al. 2022

206 – cite

279-302 – these three paragraphs can probably be shortened into one. If not then 279-297 can be at least.

312-316 – in addition to saying “it can’t be done” I would recommend emphasizing that your results are strong enough to compensate for this uncertainty.

319-324 – this needs to be much earlier. It is important to clarify early on

481 – the pough paper you cite is outdated. Use the Meiri 2010 paper you also cite in this section. You should use clade specific equations provided in the paper when possible.

539 – fig 6-8 should all be log transformed, preferably log10 transformed.

603-610 – It is unclear if species that you attribute to the genus level in your data are all the same species or not. You should clarify this in the main text. Potentially you can also check how many species exist within a genus and if there is only one write that you assume it’s that species. If not, I think that you should show an analysis of uncertainty within genera to assure readers that this did not alter results. For example, you can show that SVL varied within a genus similarly to how it varied within recognized species

624-625 – you’ve mentioned this several times now. I’d recommend removing previous references to this and leaving it here.

640-750 – Glyptosaurinae diet should be one paragraph not several. If you must then move some of it to the supplementary material.

802-814 – should be one paragraph

Pough, F.H., 1980. The advantages of ectothermy for tetrapods. The American Naturalist, 115(1), pp.92-112.

Woolley, C., Thompson, J., Wu, Y., Bottjer, D., & Smith, N. (2022). A biased fossil record can preserve reliable phylogenetic signal. Paleobiology, 1-16. doi:10.1017/pab.2021.45

Meiri, S., 2010. Length–weight allometries in lizards. Journal of Zoology, 281(3), pp.218-226.

6. PLOS authors have the option to publish the peer review history of their article (what does this mean?). If published, this will include your full peer review and any attached files.

Reviewer #1: **Yes: **Simon G Scarpetta

Reviewer #2: No

---

## [Author Response · Author response to Decision Letter 0]

7 Oct 2022

Below is a list of the changes that I made.

In response to the Journal Requirements:

1. I reformatted the title page in exact accordance with the PLOS One formatting requirements. I also double-checked that the rest of the manuscript is formatted correctly. 

2. Specimen numbers and complete repository information, including museum/institution abbreviations and geographic information, are included in S1 and S2 Datasets. The list of institutional abbreviations and full institution names are listed in the manuscript under the heading “Institutional Abbreviations.” No permits were required for this study, and I added the relevant language to the end of the “Data Collection” section under “Materials & Methods.”

3. The map in Figure 1 (previously Fig 1) was obtained from Natural Earth, which makes all its content freely available under a Public Domain CC license. According to the text on this Terms of Use page, no permission is required to use Natural Earth maps for any purpose, including commercial use, so the permission form was unnecessary. I added the necessary attribution to the Fig 2 caption as indicated in the Editor’s comments. I also embellished the map in Fig 2.

In response to the Additional Editor Comments:

• This manuscript was indeed a chapter of my dissertation. I had a feeling that the manuscript might be a bit too long, but I was not sure which material to omit, so I appreciate all the constructive suggestions about which material to rework and/or move to a separate file. I have now moved a considerable amount of the Materials & Methods content to a new file of Supplementary Information.

• I made the writing style of the manuscript sound more impersonal (e.g., removing all use of the word “my”) while being careful to not use passive voice too often. 

• I revised Table 1 and added the number of fossil species sampled per taxonomic group.

• I added a new table with regression function equations and statistics (slope, intercept, confidence intervals, SEE, etc.; see Table 3). 

• I updated the regression equation methods according to the following suggestions:

o I used the anguid regressions to estimate SVL for large-bodied fossil anguid specimens, not just the small- to medium-bodied ones. I no longer use any “reference” anguid specimens to estimate head length for large-bodied fossil anguids. 

I omitted the Helodermatidae data and regressions accordingly. I only used the largest helodermatid specimen as an additional test specimen for the anguid regressions since it is the closest morphological analogue to large glyptosaurines in my dataset. The anguid regressions proved comparably accurate in predicting the SVL of the largest helodermatid test specimen compared to the smaller extant and extinct anguid test specimens. 

I omitted the former S8 Table (Helodermatidae Regressions), S9 Table (Anguid Anatomical Ratios), and S10 Table (Anguid Reference Specimens), as they were no longer relevant.

o I ensured that no juveniles were included in any of the regression datasets.

o I used natural log to transform all the regressions for consistency. 

Natural log is commonly used to transform data in studies of body size, especially for vertebrates (e.g., Head et al 2013). Although some of the studies the editor recommended use log base 10 (e.g., Figuerido et al. 2011), several of the papers the editor recommended use natural log (Alroy 1998, 2000; Egi 2003). I also tried applying both natural log and log10 equations on some of the largest test specimens examined in this study (e.g., extant anguid equations applied to Gerrhonotus) and the resulting SVL predictions were exactly equal. Therefore, I used natural log throughout the revised manuscript. I provide this explanation in the revised methods section (Line 322). 

o I ranked the regression equations in order of preference according to the standard error of the estimate, as in Egi (2001). I updated the “Protocol for Estimating Fossil Lizard Body Size” accordingly. I also updated all body size/mass distribution and rarefaction figures accordingly. 

• I tried calculating general regression equations for all the anatomical elements I was using in order to perhaps better accommodate the largest fossil specimens. However, none of the general equations passed tests for both normality and homoskedasticity, so I was not able to use any of them. I address this in Results & Discussion. 

• I restructured the Results & Discussion section in the style of Egi (2001), with a new discussion of the accuracy of the regression methods in terms of SEE, followed by a distilled discussion of lizard body size estimates.

• I omitted the discussion of ecological and dietary adaptations in fossil lizards in the Discussion section, since the editor felt that the subject did not fit because it was not statistically quantified. I now agree that this subject was not essential to the focus of the paper. 

• Instead, I now comment more on patterns of body size diversification through time, possible sources of bias in the fossil record, and competition and temperature (using data from Zachos et al. 2008) as possible drivers of body size evolution in lizards. I include correlation graphs and statistics as additional supplementary figures to augment this discussion.

o These new figures (S5 and S6 Figs), as well as an updated version of S1 Fig (previously Fig 5) are presented as composite figures with their default formatting retained. I did not fully format these figures because I do not yet know if you will want to keep these figures. If you do think they are worth retaining, I can fully format them to make the numbers on the axes more visible (see S3 and S4 Figs, which I fully formatted because you mentioned them favorably in your comments). 

• I added a figure illustrating how I measured SVL from skeletons (the second caudal vertebra is the approximate position of the middle of the cloaca; see the new Fig 1) and cited it in the Methods and Supplementary Information.

• I added symbols to Fig 2 indicating the exact location of the fossil localities sampled. I even created a legend to categorize the localities by the lizard family sampled. Locality names, coordinates, and ages are provided with the fossil data in S1 Dataset. 

• I revised the Introduction to omit any repeated information. 

In response to the line items under Additional Editor Comments:

• I distilled the Introduction and removed the repetitive material about previous studies of fossil lizard body size.

• Line 98-99: how many families are you analyzing among extant and fossil species? Are you testing that regression slopes will not change between families? Make this question a bit clearer. Are the fossil crown lizard completely extinct taxonomic groups or can they be linked to extant families in some way?

o I added additional context to help clarify the hypothesis, including the following language: “I test this hypothesis using eight taxonomic lizard groups (seven families + Iguania, which encompasses the polytomy of iguanian families) that are represented in the Paleogene fossil record and still exist today.”

• In Table 1 you can avoid putting Ln and just mention in the caption that all measurements were natural log transformed. If for some you did not do it, you need to explain why. It will make more sense to generate another table. You need to arrange this info in a better way so that for each taxonomic group we can understand how many anatomical elements were regressed [e.g. each anatomical element can be a separate column].

o I updated Table 1 accordingly.

• Line 178-202 looks more of supporting information not needed at this stage. Try to make condense the crucial points in a small sentence and transfer these details in the Supplementary.

o I shortened this section and moved the details to the new Supplementary Information file. 

• Line 209-213: If you included both skeletonized and wet SVL estimates in the same dataset did you make sure that they correspond and correlates well in specimens where you can verify this? Using X-ray of wet-specimens you can measure the SVL based on the skeleton in a subsample [e.g. 10 random individuals] and verify if it correlates well with measurements taken from external body.

o I do not have the funding available to CT-scan a sample of wet specimens. But I carefully aligned the pelvis of a wet specimen from each of several different families with those of a dry skeletonized specimen of comparable size from each of the same families to figure out where the cloaca would be positioned on the skeleton. This is how I determined that the approximate middle of the cloaca corresponded to the posterior centrum of the second caudal vertebra. I added this clarification to the text, and I moved that paragraph to the Supplementary Information. 

• Line 217-22-: this can be another source of error because depending on the photos you might get biased measurements. Can you assure us that this is not the case? Do you have species where this can be verified (e.g. species averages SVL from wet measured individuals correspond well with those from photos of website?)?

o I did go back and measure some of these specimens again and confirmed that the measurements I had taken from photographs were accurate with a margin of error of about 5% or less (1-3 mm). I am therefore confident in my measurements. I moved this detail to the Supplementary Information and added the additional explanation. I had also forgotten to point out that some of my fossil specimens (those from UMMP) were also measured from photographs. I now mention this in the same section in the Supplementary Information. 

• Line 312 onward: this is fine although it will be more re-assuring to test if your slope do not differ between sexed individuals in a model: body_size~bone_L+sex+bone_L*sex. Same should be done for juveniles whose data points might alter sensibly the slope of only adult regression. If that is an improvement on one side, on the other it might be an additional source of error so be careful.

o I cannot perform this test because I do not have sex data for enough of the extant specimens that I measured (sex was not indicated for most of them). I eliminated all juvenile specimens from my datasets, so the latter suggestion need not apply.

• Line 806-808: this sentence is kind of suggesting us that your study is not that important. It is good that you acknowledge previous work on diversity pattern but I presume your work is a way forward. You need to re-phrase this work and make consideration about why your study is important! Also in the next section you are making a lot of justifications for your choice…this is not necessary. You need to re-calibrate your writing here.

o I moved all of the content that was under “Considerations” to the relevant parts of the Materials & Methods section or to the Supplementary Information. 

In response to Reviewer 1’s comments:

• For the fossil lizard specimens that had been previously referred to Gerrhonotinae, I updated the identification to “Anguidae indet.” and cited the references that this reviewer suggested (Ledesma et al. 2021, Scarpetta et al. 2021). 

• I changed the terms “fossil taxa” and “fossil members” to “extinct taxa” and “extinct members” throughout the manuscript. 

• I followed every individual suggestion outlined in the additional notes, with the following exception:

o “Line 354-355: I would put “Celestus” as the genus outside of the parentheses for both of these; Comptus and Siderolamprus are not currently recognized to my knowledge”

I double-checked the Reptile Database: Celestus is now the currently recognized genus over Comptus for C. c. crusculus, but Siderolamprus (Celestus) enneagrammus is still currently valid.

In response to Reviewer 2’s comments:

• I updated the SVL distribution figure (see the new Fig 6) according to the new methods employed here and with all measurements transformed using natural log. 

• I added clarification on the number of families, genera, and species analyzed in the main text under “Data Collection.”

• I recalculated lizard body mass using the family-specific and general equations from Meiri (2010). See the updated Methods section and Supplementary Information for details. 

• I commented on preservation bias in the fossil record in the new Results & Discussion section and in the Supplementary Information under “Data Collection.” I also clarified that the omitted outliers were exceptionally large in the section on “Testing and applying regressions.”

• I added new data and discussion about body size changes in the context of temperature changes through the Paleogene.

• All body size figures are now natural log transformed. 

• I turned the “Protocol for estimating body size of fossil lizards” into a flowchart. It is now a supplemental figure. 

• I followed every individual suggestion outlined in the reviewer’s notes, with the following minor exceptions:

• “Abstract – should be one paragraph”

o I don't think this matters, as I have seen plenty of abstracts with multiple paragraphs. I felt that breaking up the abstract into two paragraphs here signals the start of a new idea and makes it easier to read. 

• “206 – cite”

o Rather than providing a citation, I added an explanation of the method I used to determine the position of the cloaca on a dry skeletonized lizard specimen.

---

## [Decision Letter · Decision Letter 1]

18 Nov 2022

PONE-D-22-04244R1Body size estimation from isolated fossil bones reveals deep time evolutionary trends for lizardsPLOS ONE

Dear Dr. ElShafie,

Thank you for submitting your manuscript to PLOS ONE. After careful consideration, we feel that it has merit but does not fully meet PLOS ONE’s publication criteria as it currently stands. Therefore, we invite you to submit a revised version of the manuscript that addresses the points raised during the review process.

Modify the text accordingly following recommendations of all the reviewers.Write in a more impersonal style and reduce sensibly the descriptive section and statistical procedures in the Supporting material.Provide novel equations for reconstruction of body mass based on individual bone elements [averaged by species]. This will be a massive improvement from previous approach when SVL is necessary. Eventually you can update the body mass estimates based on more novel equations from SVL [as advised by reviewer 1] and compare them with that obtained directly from individual bone elements. Separate Results from Discussion.Provide pattern of SVL and Body mass change through time in comparison with taxonomic turnover eventually using not only the Maximum SVL (and/or body mass) but also the average values. This should be presented also for each family.Test only the hypothesis of climate impacting body size data. Biotic factors are too complex to be included in this study [even if reviewer 1 wants to see this part expanded] and they can be eventually only discussed based on visual patterns of size variation through time. ==============================

We look forward to receiving your revised manuscript.

Kind regards,

Carlo Meloro

Academic Editor

PLOS ONE

Additional Editor Comments (if provided):

I did a good job and ow the paper is more straightforward. However, I think results and discussion should be separate sections. I also recommend you to test relationship between individual bone length vs body mass. This remove the unnecessary step of computing SVL and then body mass based on other equations. Some supporting material should be condensed and some meticulous description of the study removed. The section about "Competition" is quite speculative and based on a test that uses four data points. This could be clearly the result of an artefact so please try to present only patterns of SVL through time by families and eventually explaining them in the discussion, but your data at the moment cannot clearly test this hypothesis on biotic interaction with rigour [you should account for the evolution of lizard's predators and prey as well].

Below some detailed advises for each section that I hope you will find useful.

Please try to be more impersonal in the writing and a bit more telegraphic in the description of stats test or specimen choice to avoid unnecessary long sections.

line 47: change "understand" with "investigate"

line 66: it would be good to mention base don the crocodile study what bone was considered as better predictor of mass and why..is it the femur?

line 96-129: the past tense here would be more appropriate...you tested, you sampled and so on.

Table 1: how is it possible that you sampled 3 fossil Teiidae species but no fossil species? where they all fossils of extant species or perhaps unidentified species?

Table 2 is a bit broad and I think it would be better a figure showing specimens and /or species sampled by group / time period in comparison with diversity expected as by the palaeodb database

Table 3 should be simplified showing slope as parameter (b), confidence interval and intercept + R2 and SEE. An example can be found here: https://bmcbiol.biomedcentral.com/articles/10.1186/1741-7007-10-60/tables/1

Rather than using the suffix "Element" that could be predictor and use the variable named (e.g. Skull Length and so on).

Line 230-231: I assume this large specimen was NOT larger than your extant species range. If it was the estimate will be problematic and you might need to use more general equations as the one in that Campione used for dinosaurs....basically you want to avoid making prediction of fossil species outside the range of variation of living taxa because statistically it would not be correct.

Line 242-244: What if you instead just try to test if your individual measurements can be directly used to predict body mass as well? In that case you need to average measurement by species (and/or sex) and generate your own equations to avoid the step of predicting SVL first and then body mass afterwards...it would be a much better and straightforward approach with an error we can be aware of....by predicting SVL you first generate and error that gets again into another equation to generate another form of error. Average your measurements per species and use them as predictors of body mass...you can eventually compare their SEE and their estimates with that obtained applying Meiri's equation to your SVL estimates.

Line 261: change "strong" with "consistently accurate"

Line 264-269: ok but perhaps this might be due to some outliers or some groups whose body bauplan differ from the "general" lizard one. What if you try exclusion of some groups?

Line 279: The maximum SVL is a good proxy but eventually also the weighted average or geometric mean might provide better patterns of body mass change thorugh time.

Line 309-335: this section is too long and too much descriptive. Try to cut it explaining the general pattern and the bias of the fossil record [a simple correlation test between maximum SVL [or average SVL] vs specimen number / time interval might also provide clearer pattern of bias].

Line 337-380: this section is also too long and descriptive and should eventually go before the pattern of maximum SVL through time. You first describe taxonomic richness and then after that describe the pattern of changes in SVL]. In this regard my previous advise is even more relevant...present the data from the palaeodb if you really want to compare the "diversity" you sampled with that recorded by the fossil record with pattern of genus diversity / time bin as presented in palaeodb vs the one generated based on your data. Eventually these values of species diversity can be correlated with SVL [maximum and average] to define a bias in your sample [we expect well sampled periods to show higher SVL, if taphonomic bias occurs]. By the way, this makes the study going a bit out of focus so I advise to reduce this part as much as you can in the results and eventually do another discussion section.

Sections: Competition and Climate. These mix again results with Discussion...I found this pattern a bit confusing for the reader especially because the part about the equations instead were much more like a result section. So I would be a bit more organised here and separate the "Results" from the "discussion". The competition section also is weakly supported by the data -see comments on Fig. S5 - 6, you have few data points and due to sampling bias it is hard to validate a pattern based on four data points. In theory this should be tested rigorously correcting for temporal autocorrelation so I suggest just to retain the part on climate and perhaps discuss only patterns of maximum SVL by families based on a separate supporting figure showing maximum SVL by time and family through time.

In Results you report equations, values, patterns of SVL through time and how it correlates between clades [competition] and with temperature. Present only the results plainly explaining the patterns. Then you can write a better discussion.

Line 872-879: too much details not necessary for the reader, please remove. Try to condense also previous part when unnecessary long sentence were included (e.g. line 818-827, you could just write: "For each group I aimed to sample at least 20 individuals. Only in Xenosauidae I got smaller sample size so results for this group should be considered with caution").

I can see that you got really god intention here and want to offer as many details as possible, but this makes the SI too long and hard to read in one go.

Line 884-890: same here....reduce, this is common knowledge so just mention that you tested for homoscedasticity and normality of residuals to validate assumption of the regression.

Same applies to the rest of this section. Reduce extensively also the part on sex (line 936-948). You did not have sex to test so all you are saying is quite speculative.

Line 949-958: this is a repetition not necessary and your recommendation are not based on actual test. If you provided the data, than anyone can try themself.

Line 959-985: this is too long and too detailed. Again cut and condense mentioning simply the reason why some taxa were not sampled for these groups.

Line 987-1000: if you follow my advise this bit will be not necessary. Try to generate equations based on limb elements to predict body mass. Eventually you can compare these with body mass estimates based on the application of Meiri's on your SVL

Figure S2 and the recommended procedure for generating equation is not that necessary. This is a standard procedure, so perhaps just retain Fig. S2 and remove the text associated to it in the caption (1037-1056) that is too long and repetitive.

Figure S5 and S6: I think you do not have sufficient data to test competition so remove this. It is only four families and the pattern is quite uncertain

Reviewers' comments:

Reviewer's Responses to Questions

**Comments to the Author**

1. If the authors have adequately addressed your comments raised in a previous round of review and you feel that this manuscript is now acceptable for publication, you may indicate that here to bypass the “Comments to the Author” section, enter your conflict of interest statement in the “Confidential to Editor” section, and submit your "Accept" recommendation.

Reviewer #1: (No Response)

Reviewer #2: (No Response)

2. Is the manuscript technically sound, and do the data support the conclusions?

Reviewer #1: Yes

Reviewer #2: Partly

3. Has the statistical analysis been performed appropriately and rigorously? 

Reviewer #1: Yes

Reviewer #2: Yes

4. Have the authors made all data underlying the findings in their manuscript fully available?

Reviewer #1: Yes

Reviewer #2: Yes

5. Is the manuscript presented in an intelligible fashion and written in standard English?

Reviewer #1: Yes

Reviewer #2: Yes

6. Review Comments to the Author

Reviewer #1: Dear author,

Thank you for submitting this revision! Almost all changes I requested were implemented, and the same is the case for comments from the editor and other reviewer. The manuscript is in much better shape after being shortened/condensed, having some superfluous sections removed (eg glyptosaurine diet), and having some changes made to the analyses. I have a few minor suggestions, but no major changes to request. Congratulations!

Simon Scarpetta

Minor changes

Line 65 page 4. “The first question to consider here is which variable would be best to use as a measure of body size.”

Phrasing is awkward in my opinion, I would say “It is not unambiguously clear which variable is the best proxy for body size.” Or something like that

Line 70 page 4. Add comma after “fossil vertebrates” and “many” before “fossil vertebrates”

Line 71 page 4. “Body length is often more feasible to estimate for fossil reptiles”

Relative to mammals? Dinosaurs? If the latter, have to rephrase given that dinos are a reptile

Line 101 page 5. “I use this geotemporal system as a paleontological framework”

Understand intent but this is confusing to me- maybe “I use the timeframe as a study system” or something

Line 103 page 5. Use “dactyloids” instead of “polychrotids”

Line 120 page 6. Not sure I would say polytomy, as even among pleurodontans some of the relationships are reasonably clear. So might remove this word and the of after, or clarify that not all the relationships are clear rather than saying polytomy which implies that none of them are.

Line 143 page 7. TMM. I believe they prefer TxVP (Texas Vertebrate Paleontology) now; I would confirm with Chris Sagebiel

Line 166 page 9. “anguid” should really be “anguimorph” because Heloderma is not an anguid but is presented as a modern analogue

Line 179 page 9. Delete “(Varanidae)”

Line 184 page 9. Change “Iguanidae; Polychrotidae” to “Iguania; Polychrotidae”

Line 322 page 19. Add likely between words “taxa exceeded”

Line 354 page 20. After “paleocommunities” add “ similar to modern ecosystems”

Line 359 page 21. “rarefaction analysis in S4 Fig would indicate that this study” change to . “rarefaction analysis (Fig S4) may indicate that this study”

Same line. “diversity”. Taxonomic diversity? Body size diversity? Next sentence seems to indicate body size, so please clarify here.

Line 378 page 21. “iguanians” clarify that the iguanians that occurred there then were extirpated, not iguanians in general (still plenty of iguanians in west/central NA!!)

Line 402-404. “abundant group in this study that included large-bodied individuals, negatively correlated with maximum body size in Xantusiidae and Xenosauridae, two of the most common small-bodied groups”

Something is incomplete/missing in this sentence, please clarify

Line 965 page 49. I would use “Leiolepidinae” instead of “Leiolepididae”

Line 979, page 50. Change “But” to “However,”

Line 1027, page 52. Change “recover” to “incorporate” and remove “as a result” on the next line

Reviewer #2: Overall, the ms has improved greatly and I highly recommend its publication after major revisions. I think that the drivers of body size evolution and taxonomic turnover sections can be expanded and the author should give the space necessary to properly synthesize all the data acquired and put these species in a context of living and extinct squamates. Developing allometric equations is relatively straightforward (though the amount of work done here is impressive!) I think that what would benefit the scientific community most is to better understand your interpretation of the data, and what the next steps should be moving forward in this field. Here are some things to consider adding to the section: drivers of reptile body size evolution (perhaps you can look at studies done by Meiri), maximum body size in squamates in history and how they got so big (Titanoboa, Megalania, mosasaurs), diet (unless all species here were carnivores? Herbivores are usually larger, you can check diet in Lafuma 2021), substrate use and ecology (were all species terrestrial, semi-arboreal, semi-aquatic, could this have affected body size), filling empty niches after the kpg extinction (a huge point that wasn’t mentioned), the PETM. Body size evolution is a rich field with much to dig into and I feel that it needs to be done.

-Jacob Dembitzer

General comments

I made a mistake recommending Meiri 2010 in my last review for body mass estimates. I meant Feldman 2016 (which Meiri is on) supplementary appendix 2 (see below). It has all of the regression equations needed for your study including the families missing at the moment. There is an equation for legged anuids and legless anguids. If I’m not mistaken there are a few species that you are not sure if they had limbs or not? In which case you might want to use the Anguimorpha equation, or simply assume that they had limbs if there are no known limbless glyptosaurs. In any case you definitely need to change this.

You should clarify how you measured skull length. Was it the absolute farthest point of the skull to tip of snout (in some cases that’s the jaw)? Was it to the occipital condyles?

Specific comments

27 – you say “large dataset” of fossil lizards but I think that you should be specific in saying what percent of Midwest species you studied. You can check this on the paleobiology database I think.

36 – You need to say in the study period or something as it reads a bit confusing. lizards did not reach maximum body size in the Paleogene. Pleistocene Megalania was much larger and Mosasaurs were much larger still.

40 – take out “reasonable”, estimates is good enough

49 – 50 – instead of saying “lizards occupy a wide range of…” which you say in the previous sentence, I would say “lizards (or all squamates?) make up the majority of tetrapods in harsh environments, especially deserts,” and cite Roll et al. 2017.

53 – I think you can take out unfortunately complete fossils are rare. Just say isolated elements are most common (this is true for any group though).

65 – 77 – instead of discussing crocs I would discuss how people have recently been using raw jaw length as a measure of body size across all lepidosaurs, see Herrera-Flores 2022 and 2021. Or discuss both but lepidosaurs are more relevant.

155 – and how many recognized species? Just say not all specimens that were attributed to a genus could be attributed to a specific species.

162 – maybe add a column for genera then? It’s really important that it is clear to everyone what data you collected.

187 – table 2, perhaps you could add in parentheses number of genera in each family in each NALMA

218 – log10 will make better graphs because you can label your axes much more easily (10mm, 100mm, 1000mm). if it takes too much time don’t bother as it’s not essential (though you’re readers will appreciate it).

278 – this section is perhaps the most interesting (and if I’m not mistaken the point of the study). I would expand on it either here or in the discussion I would recommend that you try to provide an overview of species' body mass evolution in squamates and other groups at this time in earth’s history. So maybe mention larger species in other places on earth like Barbaturex morrisoni and at some point, I would add a sentence about what the largest lizards in history were and compare them (mosasaurs and Megalania). You could also show the entire body size distribution of lizards in your study period in a histogram and compare them to the distribution in Feldman 2016. Or maybe just write a sentence on it. If you don’t want to add another figure then write in a sentence or table the mean, median and range for all species in the study period and then different NALMA periods. I would show body size distribution of species for the entire study period in a histogram and in the supplementary material divide it by NALMA. You could potentially do these things with both svl and mass. You can be creative with this, but I think that it should be clear that this is the meat of the research and should be expanded on. In theory you could’ve just written a paper using the allometric equations, but you’re in too deep now (and that’s a good thing).

334 – mention Megalania size

361 – its more likely that there were smaller species in the system than larger ones as larger ones fossilize better

385 – 389 – you need to be more specific; I don’t understand any of this. Why would lizards be smaller in the Paleogene and bigger in the Oligocene for the same reason? This is the interesting part, elaborate!

400 – you can remove the first paragraph of this section, it says the same thing. Or combine the two.

408 – I would add the example of Titanoboa to this section and how it was supposedly able to reach giant size because of the PETM

414-421 – really interesting analysis, you should use family as an additional variable in addition to doing it across all species. Also, how did you try to correlate them? Linear regression model? ANOVA based on each NALMA? What was the predictor and what was the response? This remains very unclear in the text. You should provide a quantitative analysis, perhaps similar to Smith 2018. Or at the very least, a very in-depth qualitative analysis/discussion. Quantitative is always better, so I would recommend that. I would personally recommend using linear regressions with estimated body size of species as the response, and temperature per million years as the predictor (check where smith got the data, I think Zachos something). You can also use family as an additional covariate.

Feldman, A., Sabath, N., Pyron, R.A., Mayrose, I. and Meiri, S., 2016. Body sizes and diversification rates of lizards, snakes, amphisbaenians and the tuatara. Global Ecology and Biogeography, 25(2), pp.187-197.

Roll, U., Feldman, A., Novosolov, M., Allison, A., Bauer, A.M., Bernard, R., Böhm, M., Castro-Herrera, F., Chirio, L., Collen, B. and Colli, G.R., 2017. The global distribution of tetrapods reveals a need for targeted reptile conservation. Nature ecology & evolution, 1(11), pp.1677-1682.

Herrera‐Flores, J.A., Elsler, A., Stubbs, T.L. and Benton, M.J., 2022. Slow and fast evolutionary rates in the history of lepidosaurs. Palaeontology, 65(1), p.e12579.

Herrera‐Flores, J.A., Elsler, A., Stubbs, T.L. and Benton, M.J., 2022. Slow and fast evolutionary rates in the history of lepidosaurs. Palaeontology, 65(1), p.e12579.

Herrera-Flores, J.A., Stubbs, T.L. and Benton, M.J., 2021. Ecomorphological diversification of squamates in the Cretaceous. Royal Society Open Science, 8(3), p.201961.

Smith, F.A., Elliott Smith, R.E., Lyons, S.K. and Payne, J.L., 2018. Body size downgrading of mammals over the late Quaternary. Science, 360(6386), pp.310-313.

7. PLOS authors have the option to publish the peer review history of their article (what does this mean?). If published, this will include your full peer review and any attached files.

Reviewer #1: **Yes: **Simon G Scarpetta

Reviewer #2: No

---

## [Author Response · Author response to Decision Letter 1]

1 Mar 2023

February 28, 2023

Dear PLoS One Editorial Board,

Thank you for the opportunity to make these additional revisions to my manuscript. Below is a list of the new changes that I made.

Overall Changes Made:

• I revised the whole manuscript to be in third-person passive voice to make it sound more impersonal.

• I separated the Results and Discussion sections and substantially revised both sections based on the Editor and Reviewer comments, including omitting the section on Competition from the Discussion. 

Body Mass Estimations and Equations

• I agree that adding equations for estimating mass of fossil specimens from individual bone measurements is useful. However, I cannot generate novel equations for body mass from individual lengths here because almost none of the extant lizard specimens that I measured included measurements of body mass. Unfortunately, lizard specimens that are dry skeletonized almost never have the original weight recorded before they are processed. Furthermore, measurements of individual cranial and limb bones are not provided in Feldman et al. (2016) or Meiri (2010), both of which were recommended by Reviewer 2 as sources for extant lizard measurements and equations for lizard body mass from SVL. Therefore, I do not have the data necessary to generate novel equations for estimating body mass from individual cranial or limb bones. 

o Instead, I translated the original equations that I provided here for estimating SVL from individual bones (Tables S1-S7) into the equations for estimating mass from SVL provided in Feldman et al. (2016; see S8 Table). The resulting equations have been added to Tables S1-S7. These translated equations can now be used to estimate lizard body mass from individual cranial and limb bones.

o The mass estimates are the same whether you calculate mass from SVL using the equations from Feldman et al. (2016) or the new equations for body mass from individual elements provided here. 

• I am still including the original equations I generated for estimating SVL from individual bones because SVL is also an important variable for paleontologists and zoologists. 

• Calculating body mass equations for individual fossil species would not be practical because fossil specimens usually cannot be identified to individual species. In addition, even genus-level assignments often change for fossil specimens upon later reexamination. 

o It is safer and more practical to develop family-level equations - that way, the body mass estimate becomes resilient to any specific taxonomic identification changes. Furthermore, family-level equations are based on a larger sample size. This approach is consistent with the rest of the paper, and with the sources used (Meiri 2010; Meiri et al. 2013; Feldman et al. 2016), which focus on family-level analyses.

New Figures and Tables

• I added the following figures:

o Fig 7. Body mass distribution by taxonomic group for fossil lizards in the Western Interior of North America through the Paleogene, including a line for the overall mean lizard body mass per NALMA.

o S6 Fig. Mean and maximum snout-vent length in fossil lizards by taxonomic group per NALMA.

o S7 Fig. Mean and maximum body mass in fossil lizards by taxonomic group per NALMA.

I added commentary about the latter two new figures in the revised Discussion as well.

• I also added a line for the overall mean lizard snout-vent length per NALMA to the SVL distribution graph in Fig 6.

• I added the following tables:

o S8 Table. Published equations for estimating lizard body mass from snout-vent length.

o S11 Table. Mean and maximum fossil lizard snout-vent length and body mass by taxonomic group per NALMA.

• The numbers of all other figures and tables were also updated accordingly. 

I made all changes suggested by the Editor and Reviewers, with the following notes…

Additional Editor Comments:

• Table 2 is a bit broad and I think it would be better a figure showing specimens and /or species sampled by group / time period in comparison with diversity expected as by the palaeodb database.

o I still prefer these data presented as a table, and I think it’s helpful to have the NALMAs listed here in table format with their age ranges. But I took the suggestion of adding the number of genera and species sampled per taxonomic group per NALMA compared to the numbers expected based on the Paleobiology Database (PBDB). These data have been added to the table. I think the result is more a reflection of how much is missing from the PBDB, but the data are now provided for comparison.

• Table 3 should be simplified showing slope as parameter (b), confidence interval and intercept + R2 and SEE.

o I made these changes, but for the sake of space and simplicity, I listed the standard error value rather than listing the full confidence interval (e.g., - X.XXX to X.XXX).

• Line 264-269: ok but perhaps this might be due to some outliers or some groups whose body bauplan differ from the "general" lizard one. What if you try exclusion of some groups?

o I did try excluding some groups, and the attempted general regressions still did not pass the tests for homoskedasticity and normal distribution of residuals. In any case, now that body mass estimation equations specific to taxonomic group are used in this study, I think a generalized lizard SVL equation is unnecessary. 

• Reduce extensively also the part on sex (line 936-948). You did not have sex to test so all you are saying is quite speculative.

o I think the comment made here is an important point to make for paleontological studies, and Reviewer 1 agreed. However, I did shorten this paragraph. 

Reviewer 2 Comments:

• I made a mistake recommending Meiri 2010 in my last review for body mass estimates. I meant Feldman 2016 (which Meiri is on) supplementary appendix 2 (see below). It has all of the regression equations needed for your study including the families missing at the moment. There is an equation for legged anuids and legless anguids. If I’m not mistaken there are a few species that you are not sure if they had limbs or not? In which case you might want to use the Anguimorpha equation, or simply assume that they had limbs if there are no known limbless glyptosaurs. In any case you definitely need to change this.

o I used updated equations from Feldman et al. (2016). The equations listed in this reference were only different for Anguidae, Xenosauridae, Iguania, and Anguimorpha; the rest were taken from Meiri (2010), so I didn’t need to change those. I used the Anguimorpha equation for Shinisauridae since an equation for Shinisauridae was not provided and Shinisauridae falls within Anguimorpha. I used the equation for Fully Legged Anguidae for the fossil anguid lizards because limb material has been found with many fossil anguid specimens – none have been interpreted as limbless. 

• 27 – you say “large dataset” of fossil lizards but I think that you should be specific in saying what percent of Midwest species you studied. You can check this on the paleobiology database I think.

o I honestly don’t know what Reviewer 2 meant by this comment. Only some of the fossil lizards in my dataset were collected in the Midwest, and I don’t know why that matters in the context of this study. If they meant that I should specify how many of the taxa in the fossil dataset still occur in the Western Interior today, I point that out in the Results section (see “Taxonomic Turnover”). I don’t think the Introduction is the place for that detail. 

• 414-421 – really interesting analysis, you should use family as an additional variable in addition to doing it across all species. Also, how did you try to correlate them? Linear regression model? ANOVA based on each NALMA? What was the predictor and what was the response? This remains very unclear in the text. You should provide a quantitative analysis, perhaps similar to Smith 2018. Or at the very least, a very in-depth qualitative analysis/discussion. Quantitative is always better, so I would recommend that. I would personally recommend using linear regressions with estimated body size of species as the response, and temperature per million years as the predictor (check where smith got the data, I think Zachos something). You can also use family as an additional covariate.

o I revised the text to clarify how I set up this regression. However, the additional analyses suggested here were not feasible because the predictor variable (lizard body size) could not be binned per million years due to the way in which the specimens were documented in museum collections (the majority of the specimens are only referred to a NALMA interval and cannot be referred to a more precise time bin). Furthermore, only two lizard families (Anguidae and Varanidae) represent the maximum lizard body size for any given NALMA, and the record of Varanidae is not continuous enough through the Paleogene to set up a correlation (see S6-S7 Fig). The correlation is already mostly a correlation between maximum anguid body size and varanid body size. 

Thank you again for the considerable time you, the Editor, and the Reviewers have contributed to reviewing this paper, and for your patience as I incorporated the suggestions into the manuscript. I look forward to hearing back from you.

Best regards,

Sara J. ElShafie, Ph.D.

Alumnus, Department of Integrative Biology and Museum of Paleontology 

University of California, Berkeley 

selshafie@berkeley.edu

---

## [Decision Letter · Decision Letter 2]

6 Apr 2023

PONE-D-22-04244R2Body size estimation from isolated fossil bones reveals deep time evolutionary trends for lizardsPLOS ONE

Dear Dr. ElShafie,

Thank you for submitting your manuscript to PLOS ONE. After careful consideration, we feel that it has merit but does not fully meet PLOS ONE’s publication criteria as it currently stands. Therefore, we invite you to submit a revised version of the manuscript that addresses the points raised during the review process.

Remove unnecessary sentences in the introduction, implement more details on some section of the methods and provide details also about the analyses of taxonomic turnover [I will be inclined to remove these anyway, it makes the paper unnecessary long and there is no clear quantification of diversity patterns and extinction/origination rates, this might require a paper on its own] and maximum body size as well as rarefaction [how the analysis was done and to test what]. Result section require several unnecessary comments to be removed and shifted in the Discussion. The discussion section requires a complete re-styling following up on the aims of the paper [discussing first the reliability of the equations and then the pattern of body size changes through time].  Figure 1 should be supplied with more graphical details on how single bones were measured. The section Appendix should be part of supporting material [Appendix S1] and needs to be more in line with PlosOne style and nomenclature (https://journals.plos.org/plosone/s/supporting-information). The other files are good but maybe S2 Fig should go back in the main results.    

We look forward to receiving your revised manuscript.

Kind regards,

Carlo Meloro

Academic Editor

PLOS ONE

Additional Editor Comments:

The paper improved considerably however the structure is still confusing. Some sections in materials and methods need to be expanded and the entire part concerning taxonomic turnover or the analyses through time of maximum body size are not described at all. Also the discussion now focuses entirely on the trends observed in the fossil species without any consideration on the equation presented, their accuracy and their validity. This paper still require a better direction. See the attached pdf comments and please check carefully the style of the paper you would like to achieve and be coherent. A guide example could be to follow the structure as in https://doi.org/10.1371/journal.pone.0082000 on how to present and discuss predicting equation. The part about temporal shift in body mass can still be implemented but must follow a better structure. Another good example to look at is: https://doi.org/10.1016/0024-4066(95)90015-2

At the moment the sections are incomplete and confusing so please try to still incorporate essential methods in the write section -and not in the appendix- while unnecessary details can remain in the Supporting Material that needs to follow the style of PlosOne [there is no SI for Methods and or Discussion, each must be numbered and valid on its own].

Reviewers' comments:

Reviewer's Responses to Questions

**Comments to the Author**

1. If the authors have adequately addressed your comments raised in a previous round of review and you feel that this manuscript is now acceptable for publication, you may indicate that here to bypass the “Comments to the Author” section, enter your conflict of interest statement in the “Confidential to Editor” section, and submit your "Accept" recommendation.

Reviewer #1: All comments have been addressed

Reviewer #3: (No Response)

2. Is the manuscript technically sound, and do the data support the conclusions?

Reviewer #1: Yes

Reviewer #3: Yes

3. Has the statistical analysis been performed appropriately and rigorously? 

Reviewer #1: Yes

Reviewer #3: Yes

4. Have the authors made all data underlying the findings in their manuscript fully available?

Reviewer #1: Yes

Reviewer #3: Yes

5. Is the manuscript presented in an intelligible fashion and written in standard English?

Reviewer #1: Yes

Reviewer #3: Yes

6. Review Comments to the Author

Reviewer #1: Dear Author,

Thanks for submitting this revision! I think the manuscript is in great shape and I am happy to recommend the paper for publication—thanks for incorporating all of my and the other reviewer’s/editor’s comments and suggestions. I have attached a handful of extra comments that you should look at (attached reviewer PDF), but otherwise all good. Congrats! I am very excited to see this paper published.

Simon Scarpetta

Reviewer #3: Dear Editor,

thank you for the opportunity to review this manuscript. It is very interesting in my opinion, however, it needs to be improved in fluency and clarity.

Introduction

The section seems to follow the conventional scheme, however, it is a bit confusing in some periods.

Rows 74-79: To make the text easier to read, I advise eliminating all the information in the brackets and just leaving the references.

Rows 130:133: I suggest do not anticipate the results at the end of the Introduction. Instead, give an overview of the argument and anticipate the methods.

Material and method

I would suggest starting the data collection section by saying the total number of collected specimens and the number of genera they represent.

Rows 227-229: I suggest putting this sentence in line with rows 232-235 sayings that although bivariate regression is frequently used, you chose to utilise RMA instead, and then discuss the variables you employed.

Row 247: I suggest clarifying how you tested the regression.

The Estimating body mass for fossils section needs to be reorganized, in my opinion. Please see the reviewed pdf for details.

Results

The section should be reorganised so that only the results description appears. Any additional comments or citations must be added to the discussion.

I would also recommend that each of the results sections begin with an introductory context. For example, you could begin the section Estimating body size of fossil lizards with "Results investigating the relationships between different skeletal elements and body size revealed..."

In rows 301-303 you say: “The results presented here also indicated that taxonomic diversity was greatest in the early Eocene and decreased in the early Oligocene”

However, based on Figures 6-7, it appears that taxonomic diversity decreased during the late Eocene and increased during the early Oligocene, although not reaching the Eocene levels. By the end of the Oligocene taxonomic diversity decreased again.

Could be also interesting to investigate the relationships between climate and taxonomic diversity. From Figures 6-7 seems that the diversity reached two picks. The first seems to occur during the warmest Eocene phase: the Paleocene – Eocene Thermal Maximum, and the second seems to coincide with the cooling phase of the Oligocene.

It is not very clear from the Materials and Methods section how you computed species turnover and the rarefaction analysis. Also in this case, describe only the diversity trend in Figures 6 and 7 and interpret the results in the Discussion.

It is not very clear from the main manuscript that the fossil dataset includes also juveniles, in my opinion. I would propose also to debate if the inclusion of these samples could affect the outcomes.

Please see the attached pdf file for minor comments.

7. PLOS authors have the option to publish the peer review history of their article (what does this mean?). If published, this will include your full peer review and any attached files.

Reviewer #1: **Yes: **Simon G Scarpetta

Reviewer #3: No

---

## [Author Response · Author response to Decision Letter 2]

6 Oct 2023

Dear PLoS One Editorial Board,

Thank you for the opportunity to make these additional revisions to my manuscript. Below is a list of the new changes that I made.

Overall Changes Made:

• I revised the Introduction to make it more concise. 

• I heavily reworked the Materials and Methods section to be sufficiently detailed and more cohesive.

• I took the text that had been moved to Supporting Information, distilled it, and incorporated it back into the main text.

• I added mean values to Table 4 to provide the full data visualized in Fig 6, Fig 7, S6 Fig, and S7 Fig. I also revised the captions for those figures. 

• I removed all material related specifically to taxonomic turnover. This included removal of the former Table 2 and S3 Fig, which were no longer relevant.

I made all changes suggested by the Editor and Reviewers, with the following minor exceptions:

Editor comments and edits (lines refer to previous version of the manuscript)

• Line 18: With respect to the Editor’s comment, I did not indicate that lizards are more vital to extant ecosystems than pollinating insects or any other group, only they that they are important. I elaborate on this with citations in the Introduction. I maintain that this is a reasonable and appropriate opening sentence for this abstract. 

• Lines 32-35: I like the more concise phrasing suggested here. However, the femur was only possible to regress for Anguidae, not the other lizard groups, so I omitted that mention. 

• Lines 159-160: I think the justification for the chosen study system is best placed in the Introduction, and does not need to be reiterated in the Materials & Methods section.

• Line 370 – “do you mean correlation r????”: I mean the coefficient of determination for the correlation, R2. I think this sentence was confusing as written, so I revised it. 

• Lines 395-406: With respect, I do not think that this information is off topic. It helps to explain why we might be seeing the observed patterns even though the maximum body sizes to not correlate with global temperature data. I left this information in the “Climate as a potential driver of body size evolution” section of the Discussion, but I did revise that section to be more cohesive. 

• The section Appendix should be part of supporting material [Appendix S1] and needs to be more in line with PlosOne style and nomenclature (https://journals.plos.org/plosone/s/supporting-information). 

o I am not sure what the Editor meant by this comment because there was no section in the manuscript labeled “Appendix.” If they were referring to the supporting datasets (S1 and S2 Datasets), I see no reason to change the file name because “Dataset” is listed as one of the common item descriptions in the Supporting Information guidelines link provided by the Editor. Those files are datasets, and I would prefer to designate them as such for clarity. Furthermore, I did not see any instructions in the linked guidelines as to a required format for dataset files. If the Editor would like to request specific changes to the formatting of S1 and S2 Datasets with explanations for the needed changes, I would be happy to make those changes. 

Reviewer #3 comments and edits (lines refer to previous version of the manuscript)

• Lines 234-238: Log was not written out as “logarithm” in the PLoS One publication that the Editor recommended as a formatting reference (Field et al. 2013), and “log” is the more common term used, so I did not see the need to write it out here. 

• Lines 267-272: I agree that the explanation of how I derived new equations for estimating body mass was not very clear as written. I revised it to be more coherent. However, I decided to stick with the existing arrangement of explaining the equations that had already been published and then explaining how I derived new equations because I had to use the existing published equations to generate the new equations. 

• Line 419: Correct, mass cannot be zero, but the mass listed here is 1kg, with a natural log of 0.00.

• I would suggest starting the data collection section by saying the total number of collected specimens and the number of genera they represent.

o I understand what the reviewer is getting at – it would be confusing to only discuss the fossil data but not the extant data before presenting a table that offers both fossil and extant data. However, this is just an artefact of the formatting for the submitted manuscript. Since the tables will be separated from the text in the final layout, this will no longer be an issue. I think the text flows better with the details for the fossil data presented together in one paragraph, followed by the details for the extant data in the next paragraph. But I also revised the entire Data Collection section to improve its clarity. 

• Rows 227-229: I suggest putting this sentence in line with rows 232-235 sayings that although bivariate regression is frequently used, you chose to utilise RMA instead, and then discuss the variables you employed.

o RMA is a form of bivariate linear regression, not an alternative to it. I added the word “bivariate” after RMA to clarify this for anyone who is not familiar with RMA regressions. 

• It is not very clear from the main manuscript that the fossil dataset includes also juveniles, in my opinion.

o There was a brief mention of this in the Methods, and the SI included commentary on why juveniles were not included in the extant dataset and why they were likely not represented in the fossil dataset. I revised that commentary and moved it back to the Methods section in the main text. 

Thank you again for the additional time that you, the Editor, and the Reviewers have contributed to reviewing this paper, and for your patience as I incorporated the suggestions into the manuscript. I look forward to hearing back from you.

---

## [Decision Letter · Decision Letter 3]

23 Oct 2023

PONE-D-22-04244R3Body size estimation from isolated fossil bones reveals deep time evolutionary trends in North American lizardsPLOS ONE

Dear Dr. ElShafie,

Thank you for submitting your manuscript to PLOS ONE. After careful consideration, we feel that it has merit but does not fully meet PLOS ONE’s publication criteria as it currently stands. Therefore, we invite you to submit a revised version of the manuscript that addresses the points raised during the review process.

These include few minor issues in the text and a more effective version of Fig. 7 and 8 which are too identical and deserve a climate curve at the bottom. Try also to move some of the results that are now in the discussion in the results section to imrpove readability and to be coherent with your Methods section where rarefaction is mentioned.  

We look forward to receiving your revised manuscript.

Kind regards,

Carlo Meloro

Academic Editor

PLOS ONE

Journal Requirements:

Additional Editor Comments:

The paper is greatly improved. I just suggest final adjustments needed especially because Fig. 7 and 8 are identical so it worth or combining them or showing this pattern with climate curve at the bottom.

Some of my additional advises are in conjunction with that of the reviewers so please try to follow them:

In line 32 write: Individual bones predict SVL of fossil lizards with high level of accuracy (R > 0.9)

Line 36: please add another sentence about the significance of this result. Something like: Maximum size decreased in late Palaeogene and no association could be found between lizard body size and proxies of climatic changes.

Line 46: change "roles" with "niches"

Line 48: after semi-colon add "they can maintain" and remove semi-colon before "and"

Line 50: change "animals" with for many more "animal species"

Line 103: as also advised by the reviewer change "extirpated" with "clades that become extinct"

Line 150: change "amassed" with "collected"

Line 152: you can remove "collected"

Line 208: here you need to clarify if your n=17 stands for number of species or number of specimens...to my understanding you mixed up specimens and species in the same dataset, am I correct? please clarify

Figure 7 and 8 are identical so maybe just add an axis to the right showing body mass values that reflect the SVL values or add this into the appendix. It will be good to see a climate Oxygen curve at the bottom of this graph. Nothing fancy but something similar to this Fig. 3: "" ext-link-type="uri" xlink:type="simple">https://www.science.org/doi/10.1126/science.1194830"

Line 469-470: this should be part of the Results and be commented upon in the discussion from 471 onwards.

Line 507-510: again this must go in the results. Conclude your Methods also mentioning that you will test the hypothesis of climatic influence on global lizard maximum, minimum and average body size using oxygen isotopes as proxy for climate. Again, in the results please report the correlation test (Pearson or Spearman, very easy to be done in PAST) because it is NOT lizard body size to predict climate, but maybe the other way round (biologically makes more sense which is also in line with your supplementary figures....based on what you wrote it seems that you wanted to predict climate from lizard body size). A simple correlation test will be sufficient with climate curve implemented at the bottom of Figure 7.

Reviewers' comments:

Reviewer's Responses to Questions

**Comments to the Author**

1. If the authors have adequately addressed your comments raised in a previous round of review and you feel that this manuscript is now acceptable for publication, you may indicate that here to bypass the “Comments to the Author” section, enter your conflict of interest statement in the “Confidential to Editor” section, and submit your "Accept" recommendation.

Reviewer #3: (No Response)

2. Is the manuscript technically sound, and do the data support the conclusions?

Reviewer #3: Yes

3. Has the statistical analysis been performed appropriately and rigorously? 

Reviewer #3: Yes

4. Have the authors made all data underlying the findings in their manuscript fully available?

Reviewer #3: Yes

5. Is the manuscript presented in an intelligible fashion and written in standard English?

Reviewer #3: Yes

6. Review Comments to the Author

Reviewer #3: Thank you for the opportunity to review this amended and enhanced version of the work! I believe that the work is in good shape, and I am pleased to recommend it for publication. Minor comments are included in the attached pdf version. I would recommend adding a new table explaining the collected measurements, in particular. I also propose that "extinct" be used instead of "extirpated" and that "collected" be used instead of "amassed." In this instance, the two words are more suitable.

7. PLOS authors have the option to publish the peer review history of their article (what does this mean?). If published, this will include your full peer review and any attached files.

Reviewer #3: No

---

## [Author Response · Author response to Decision Letter 3]

7 Dec 2023

Dear PLoS One Editorial Board,

Thank you for the opportunity to make these additional revisions to my manuscript. 

I made all changes suggested by the Editor and Reviewer #3, with the following minor exceptions:

Editor comments and edits (lines refer to previous version of the manuscript)

• In line 32 write: Individual bones predict SVL of fossil lizards with high level of accuracy (R 0.9)

o I agree with the editor that the sentence, as it was written, needed improvement. I also realized that the sentence was not conveying what I had intended. I revised it accordingly for clarity. 

• Line 46: change "roles" with "niches"

o For the term “niche,” I refer to Van Valen’s 1971 paper in which he regards “niche” as the specific ecological resource space that a species occupies [Van Valen, L. "Adaptive zones and the orders of mammals." Evolution (1971): 420-428]. Thus, for the context of the sentence in question, I consider “roles” to be a more appropriate term.

• Figure 7 and 8 are identical so maybe just add an axis to the right showing body mass values that reflect the SVL values or add this into the appendix. It will be good to see a climate Oxygen curve at the bottom of this graph. Nothing fancy but something similar to this Fig. 3: "https://www.science.org/doi/10.1126/science.1194830"

o I agree that it was a good idea to combine the SVL and mass figures since they were essentially the same. I also liked the idea of adding in the Zachos curve. In the interest of keeping the figure from looking too crowded, I accomplished this by reflecting the LN(Mass) scale onto the LN(SVL) scale on the lefthand side, and adding the Zachos Curve scale on the righthand side. I also simplified the Zachos Curve into a single dotted line representing the mean annual paleo temperature (MAPT) for the midpoint of each NALMA interval, as derived from ∂O18 isotope proxies. I noted these changes in the new Figure 7 caption.

• Line 507-510: again this must go in the results. Conclude your Methods also mentioning that you will test the hypothesis of climatic influence on global lizard maximum, minimum and average body size using oxygen isotopes as proxy for climate. Again, in the results please report the correlation test (Pearson or Spearman, very easy to be done in PAST) because it is NOT lizard body size to predict climate, but maybe the other way round (biologically makes more sense which is also in line with your supplementary figures....based on what you wrote it seems that you wanted to predict climate from lizard body size). A simple correlation test will be sufficient with climate curve implemented at the bottom of Figure 7.

o You are correct that I accidentally reversed my dependent and independent variables here – thank you so much for catching that! I corrected the text in the caption for S5 Fig. I also moved the relevant text from the Discussion to the end of both the Methods and the Results sections and revised accordingly. Furthermore, I ran correlations for Global temperature vs. Maximum and mean mass in addition to SVL. I used RMA regression correlation statistics to be consistent with the methods in the rest of the manuscript. 

Reviewer #3 comments and edits (lines refer to previous version of the manuscript)

• Also from Editor: Lines 103, 483: “change ‘were extirpated’ to ‘gone extinct’”

o The term “extirpated” means that the clade no longer occurs in that particular geographic area, but still exists elsewhere. “Extinct” means that the clade no longer exists anywhere. The lines in question refer to clades such as varanids and xenosaurids¬¬ that no longer occur in the Western Interior of North America but still exist elsewhere, so the term “extirpated” would be more appropriate than “extinct.” However, in case the Editor and Reviewer #3 think that this term would be unfamiliar for most readers, I changed the wording to “locally extinct.”

• Lines 151-152: “It would be great if you could include a table describing the anatomical measurements that were taken.”

o Table 2 already includes a list of each anatomical measurement taken for each taxonomic group, and Figure 1 visually indicates how each of those measurements were taken. I do not think that adding a table with any further description of the anatomical measurements would contribute value to the paper. On the contrary, it would be redundant and take up space. 

• Lines 202-203, 366-367: I’m not sure if the outer parentheses should be replaced by brackets here since brackets are used for citation numbers in PLoS One’s format. I will leave this up to the discretion of the copy editor. 

I also made the following revisions to figures:

• As suggested by the Editor, I combined Figures 7 and 8 into a single figure (now just Fig 7) that shows data representing both SVL and mass. I also added the Zachos curve representing global marine temperature to this figure. 

• I rearranged the layout somewhat in Figures 1 and 3 to make things less cluttered.

Thank you again for the additional time that you, the Editor, and the Reviewers have contributed to reviewing this paper, and for your patience as I incorporated the suggestions into the manuscript. I look forward to hearing back from you.

Best regards,

Sara J. ElShafie, Ph.D.

Alumnus, Department of Integrative Biology and Museum of Paleontology 

University of California, Berkeley 

selshafie@berkeley.edu

---

## [Editor Report · Decision Letter 4]

11 Dec 2023

Body size estimation from isolated fossil bones reveals deep time evolutionary trends in North American lizards

PONE-D-22-04244R4

Dear Dr. ElShafie,

We’re pleased to inform you that your manuscript has been judged scientifically suitable for publication and will be formally accepted for publication once it meets all outstanding technical requirements.

Kind regards,

Carlo Meloro

Academic Editor

PLOS ONE

Additional Editor Comments (optional):

Thanks for taking all the suggestions onboard. Well done, I hope this process was useful for you.
---

## [Editor Report · Acceptance letter]

26 Dec 2023

PONE-D-22-04244R4 

PLOS ONE

Dear Dr. ElShafie, 

I'm pleased to inform you that your manuscript has been deemed suitable for publication in PLOS ONE. Congratulations! Your manuscript is now being handed over to our production team.

Kind regards, 

on behalf of

Dr. Carlo Meloro 

Academic Editor

PLOS ONE